# Improving Classifier-Free Guidance in Masked Diffusion: Low-Dim Theoretical Insights with High-Dim Impact

**Kevin Rojas**
Georgia Institute of Technology
kevin.rojas@gatech.edu

**Ye He**
Georgia Institute of Technology
yhe367@gatech.edu

**Chieh-Hsin Lai**
Sony AI
Chieh-Hsin.Lai@sony.com

**Yuhta Takida**
Sony AI
yuta.takida@sony.com

**Yuki Mitsufuji**
Sony AI
yuhki.mitsufuji@sony.com

**Molei Tao**
Georgia Institute of Technology
mtao@gatech.edu

## Abstract

Classifier-Free Guidance (CFG) is a widely used technique for conditional generation and improving sample quality in continuous diffusion models, and its extensions to discrete diffusion has recently started to be investigated. In order to improve the algorithms in a principled way, this paper starts by analyzing the exact effect of CFG in the context of a low-dimensional masked diffusion model, with a special emphasis on the guidance schedule. Our analysis shows that high guidance early in sampling (when inputs are heavily masked) harms generation quality, while late-stage guidance improves it. These findings provide a theoretical explanation for empirical observations in recent studies on guidance schedules. The analysis also reveals an imperfection of the current CFG implementations. These implementations can unintentionally cause imbalanced transitions, such as unmasking too rapidly during the early stages of generation, which degrades the quality of the resulting samples. To address this, we draw insight from the analysis and propose a novel classifier-free guidance mechanism. Intuitively, our method smooths the transport between the data distribution and the initial (masked) distribution, resulting in improved sample quality. Remarkably, our method is achievable via a simple one-line code change. Experiments on conditional image and text generation empirically confirm the efficacy of our method.

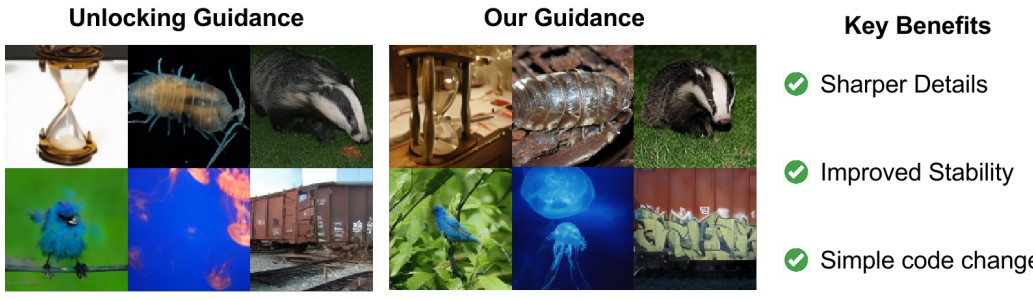

Figure 1: We proposed an improved guidance mechanism through column normalization. Our method produces sharper images while being more stable to the guidance strength. Notably, it requires only a minor code modification.

# 1 INTRODUCTION

Continuous-state diffusion models (Ho et al., 2020; Song et al., 2021) have proven effective in both unconditional and conditional generation tasks, such as generating data from natural language prompts. Prominent examples include text-to-image and text-to-video models like Stable Diffusion, Sora, and others (Rombach et al., 2022; Esser et al., 2024; Brooks et al., 2024). More recently, progress in discrete diffusion modeling (Campbell et al., 2022; Lou et al., 2023; Huang et al., 2023; Gruver et al., 2023; Ou et al., 2024; Shi et al., 2024; Sahoo et al., 2024) has extended the applicability of diffusion-based generation to new domains, including molecular design, protein synthesis, and languages.

Despite their success, these models often produce outputs that lack fine detail or strong alignment with conditioning inputs (e.g., text prompts). A widely adopted technique to address this issue is classifier-free guidance (CFG) (Ho & Salimans, 2021), which improves fidelity but typically at the cost of reduced sample diversity (Karras et al., 2024).

A growing body of work has sought to understand the theoretical foundations of CFG in diffusion models (Chidambaram et al., 2024; Pavasovic et al., 2025; Bradley & Nakkiran, 2024; Ye et al., 2025), while others have developed improved guidance algorithms (Karras et al., 2024; Li et al., 2024). Classifier-free guidance has also been adapted to discrete diffusion models (Nisonoff et al., 2024; Schiff et al., 2024), yielding promising empirical gains.

Among these improvements, dynamic guidance schedules—where guidance strength varies over the generation trajectory—have shown especially effective. Strategies such as guidance intervals (Kynkäänniemi et al., 2024) and gradually increasing schedules (Xi et al., 2024) can significantly enhance sample quality and are increasingly adopted in practice (Hoogeboom et al., 2024; Yu et al., 2024; Karras et al., 2024). However, such scheduling techniques remain exclusive to the continuous setting.

While recent adaptations of CFG to discrete diffusion have improved empirical performance, defining and optimizing effective guidance strategies in discrete spaces remains a fundamentally challenging and open research problem.

In our work we aim to better understand the mechanisms by which guidance affects the sampling process in discrete diffusion. Specifically, we aim to answer the following questions:

- How does the guidance schedule affect the distribution of the generated samples?

- Is it possible to characterize properties of good guidance schedules?

To do so, we start by deriving explicit formulas for the sampled distribution under varying guidance schedules in 1 and 2 dimensions. Our analysis not only reveals flaws in current CFG implementations, but also leads to effective design principles for effective guidance schedules in masked diffusion. Our contributions can be summarized as:

- We identify a key flaw in existing discrete guidance mechanisms that complicates simulation, and provide a theoretical explanation of its cause.

- To address the flaw, we propose a novel classifier-free guidance mechanism based on a simple yet principled column normalization of the rate matrix. This change is theoretically justified, easy to implement (pseudocode in Sec.1), and compares favorably to existing approaches in practice.

- The first theoretical justifications to characterize guidance schedules and the mechanisms by which they improve sample generation

```python
def normalized_guidance_euler_transition(
    x, c, t, dt, w
):
    uncond = model(x, cond=None)
    cond = model(x, cond=c)
    logits = w * cond + (1 - w) * uncond

    p_theta = logits.softmax(dim=-1)

    s, s_bar = sigma(t), sigma_bar(t)
    change = dt * s * (1 - exp(-s_bar))
    return sample(delta(x) + change * p_theta)
```

```python
def other_guidance_euler_transition(
    x, c, t, dt, w
):
    uncond = model(x, cond=None)
    cond = model(x, cond=c)
    logits = w * cond + (1 - w) * uncond

    p_theta = logits.exp()

    s, s_bar = sigma(t), sigma_bar(t)
    change = dt * s * (1 - exp(-s_bar))
    return sample(delta(x) + change * p_theta)
```

Listing 1: Our guidance in the special case of masked diffusion using Euler transitions. Our method is a simple one line change *but clearly motivated by theory*
.

Listing 2: Unlocking/Simple guidance for the special case of masked diffusion using Euler transitions.

## 2 PRELIMINARIES

This paper considers a vocabulary of size $M$ and state space $S = \{1, 2, \ldots, M\}^d$, with each element being a sequence of tokens. The number of tokens $d$ will also be referred to as the dimension. Each probability distribution on $S$ is represented as a vector in $\mathbb{R}^{M^d}$ whose entries sum to one.

### 2.1 INTRODUCTION TO DISCRETE DIFFUSION VIA CTMC

Given an initial distribution $p \in \mathbb{R}^{M^d}$, discrete diffusion is defined by considering a rate matrix $R_t \in \mathbb{R}^{M^d \times M^d}$ and defining a continuous time Markov chain (CTMC):

$$\frac{\mathrm{d}p_t}{\mathrm{d}t} = R_t p_t, \quad p_0 = p. \tag{1}$$

we pick $R_t$ such that when $t \to \infty$, $p_t$ converges to a simple distribution. Additionally, $R_t$ must satisfy that its non-diagonal entries are non-negative and each column must add up to zero. The time reversal of this process corresponds to a different CTMC given by:

$$\frac{\mathrm{d}p_{T-t}}{\mathrm{d}t} = \overline{R}_{T-t} p_{T-t}. \tag{2}$$

This process is considered as the time reversal since it has the same law as (1) for all values of $t$ and the reverse transition matrix can be found through the following identities:

$$\overline{R}_t(y, x) = R_t(x, y) \cdot \frac{p_t(y)}{p_t(x)}, \qquad \overline{R}_t(x, x) = -\sum_{y \neq x} \overline{R}_t(y, x). \tag{3}$$

The ratios $\frac{p_t(y)}{p_t(x)}$ are called the concrete score and they enable sampling through Euler schemes, $\tau$-leaping (Lou et al., 2023) or higher order methods (Ren et al., 2025).

**Masked Discrete Diffusion** is a special case of diffusion where a clean sequence $x_0$ is gradually corrupted over time by randomly masking some of its entries. Typically, the forward process is chosen such that at time $t = 0$, the data is completely unmasked, and at $t = T$ the data is completely masked. Formally, the distribution of each token can be written in a simple form:

$$p_t(x_t^i | x_0) = \begin{cases} x_0^i & \text{with probability } e^{-\overline{\sigma}_t} \\ M & \text{with probability } 1 - e^{-\overline{\sigma}_t} \end{cases}$$

Where $\overline{\sigma}_t$ is an increasing function that defines the unmasking schedule. The forward dynamics are defined such that tokens transition only from a clean state to a masked state, remaining masked thereafter. Generation is achieved by starting from a fully masked state and iteratively unmasking tokens until a clean sequence is recovered by following Equation (2).

Masked diffusion enjoys a simple and structured design, which has enabled its successful scaling to large practical tasks (Nie et al., 2025; Xie et al., 2025; Ou et al., 2024; Sahoo et al., 2024; Shi et al., 2024; Campbell et al., 2022). For this reason, we adopt it as the primary setting for our analysis.

## 2.2 CLASSIFIER-FREE GUIDANCE

Classifier-free guidance (CFG) (Ho & Salimans, 2021) was introduced to improve conditional diffusion models, like generating images from class labels or text. Models often failed to capture fine details, which led to less accurate and misaligned samples (Karras et al., 2024).

CFG tackles this by comparing predictions with and without conditioning, and biasing generation toward the conditional signal. Formally, the method defines a reweighted distribution:

$$p^{(w)}(x|y) \propto p^w(x|y)p^{1-w}(x)$$

Where $w$ is called the guidance strength. Setting $w = 1$ recovers the usual conditional distribution $p(x|y)$ while $w = 0$ corresponds to unconditional sampling. The crucial insight is that by setting $w > 1$ it is possible to emphasize the conditional part, effectively pulling the generation closer to satisfying the required condition. CFG is now a standard tool in conditional diffusion models, more controllable generations across tasks such as text-to-image synthesis.

While the original formulation contrasted the conditional model against its unconditional counterpart, later works recognized that this can be extended by replacing the unconditional distribution with other distributions. For example, Karras et al. (2024) used a weaker conditional model as the guiding distribution. This view has led to the understanding that the essence of guidance lies in balancing a **target distribution** $p$ with a **guiding distribution** $q$.

$$p^{(w)}(x) \propto p^w(x)q^{1-w}(x) \tag{4}$$

This view highlights that the unconditional model is simply one possible choice of $q$. By carefully selecting $q$ recent works (Karras et al., 2024; Li et al., 2024; Rojas et al., 2025) have proposed novel guidance strategies that further improve sample quality and control.

## 2.3 GUIDANCE FOR DISCRETE DIFFUSION MODELS

In parallel to advances in continuous domains, discrete diffusion models have emerged as powerful generative models, enabling diffusion-based approaches on modalities that were previously out of reach—most notably, text. Improving the fidelity and controllability of these models is crucial, and guidance offers a natural path forward. Extending classifier-free guidance to the discrete setting has therefore become an active line of research with two main approaches having been proposed, which we describe below, followed by a discussion in Section 3.3 comparing them to our method.

**Unlocking Guidance** (Nisonoff et al., 2024) introduced the first classifier-free guidance mechanisms for discrete diffusion models. Inspired by the continuous case, they constructed a guided backwards transition by interpolating between two transition matrices in equation 2, yielding

$$\overline{R}_t^{(w)}(y,x) = R_t(x,y) \cdot \left(\frac{p_t(y)}{p_t(x)}\right)^w \left(\frac{q_t(y)}{q_t(x)}\right)^{1-w}, \qquad \overline{R}_t^{(w)}(x,x) = -\sum_{y \neq x} \overline{R}_t^{(w)}(y,x), \tag{5}$$

where $p_t, q_t$ follows the forward CTMC (1). Here $p_0 = p$ is the distribution that we want to generate from and $q$ serves as the guiding distribution. [1]. Notice how the products mimic those present in equation 4. A useful way to interpret this is by introducing the notion of the **tilted distribution**:

$$p^{(w)}(x) = Z_w^{-1} p^w(x) \cdot q^{1-w}(x), \qquad Z_w = \sum_{y \in S} p^w(x) \cdot q^{1-w}(x).$$

The generation process follows the dynamics induced by the guided transition matrix substituted in equation 2. Nisonoff et al. (2024) showed that guidance in the discrete setting serves a role analogous to its continuous counterpart—steering the model toward more faithful conditional samples—thus providing an important step toward improving the quality of discrete diffusion generations.

**Simple Guidance.** Concurrently, Schiff et al. (2024) proposed an alternative formulation of classifier-free guidance for discrete diffusion. Rather than interpolating the rate matrices as in

---

[1]In existing literature, $p$ is usually a class-conditional distribution, and $q$ is an unconditional distribution. We adopt the general setup since recent works have shown that $q$ can be chosen in different ways (Karras et al., 2024; Li et al., 2024; Rojas et al., 2025).

Nisonoff et al. (2024), they directly interpolate the transition probabilities themselves. Specifically, when transitioning from time $t$ to time $s < t$, the following transition was proposed:

$$p_{\text{simple}}^{(w)}(z_s|z_t) \propto p^w(z_s|z_t)p^{1-w}q(z_s|z_t). \tag{6}$$

As before, increasing $w$ biases towards the target distribution $p$. Although the construction appears different, in the limit $s \to t$ the transitions coincide with those of Nisonoff et al. (2024). In practice, however, a finite number of steps is used, and the resulting methods are distinct. To implement these transitions, one can use equation (2) together with a suitable numerical integration scheme.

## 2.4 DYNAMIC GUIDANCE SCHEDULES

In our work we will consider dynamic guidance schedules, i.e. making $w$ a function of time. Such schedules have become more popular in practice. For instance, guidance interval (Kynkäänniemi et al., 2024) only applies guidance on a segment of the generation process. Doing so produces a boost in the performance of diffusion models. However, existing work on dynamic guidance schedules (Kynkäänniemi et al., 2024; Xi et al., 2024) has been limited to a continuous (state-space) diffusion models. It remains unclear whether such schedules are also effective in discrete state diffusion—a question that serves as the main focus of our investigation.

Specifically, this work will consider $w : [0, T] \to \mathbb{R}$, i.e. guidance strength as a function of time, referred to as the guidance schedule. The schedule induces a generative process given by:

$$\frac{\mathrm{d}p_{T-t}}{\mathrm{d}t} = \overline{R}_{T-t}^{(w_{T-t})}p_{T-t} \tag{7}$$

Understanding which schedules result in the best generation is of crucial importance to further improve the sample accuracy of discrete diffusion models.

## 3 METHODOLOGY

We begin by analyzing the guided process in the simplest case of a single token in Section 3.1, which already reveals a key limitation of existing guidance. We then introduce our proposed remedy in Section 3.2 via column normalization. Afterwards, we analyze the effect of guidance schedules on two tokens in Section 3.4. Finally, we present experimental results of our methods in Section 4.

### 3.1 IDENTIFYING AN ISSUE IN THE GUIDANCE OF DISCRETE DIFFUSION

We start by studying guidance for $d = 1$ where an exact analysis is possible. The following result characterizes the distribution at time $t$ under constant guidance:

**Theorem 3.1.** *(Informal) Along the dynamics of equation (7), starting from a fully masked state, the distribution at time $t$ is given by:*

$$p_t = \left(1 - \left(\frac{1 - e^{-\overline{\sigma}_t}}{1 - e^{-\overline{\sigma}_T}}\right)^{Z_w}\right) \cdot p^{(w)} \tag{8}$$

We present a full proof, as well as a more general result for varying guidance schedules in Theorem B.1. This shows that for $d = 1$ the guided process exactly recovers the tilted distribution, with the unmasking speed controlled by the factor in front of $p^{(w)}$. Although low-dimensional, this result already reveals important properties of the guided backwards process.

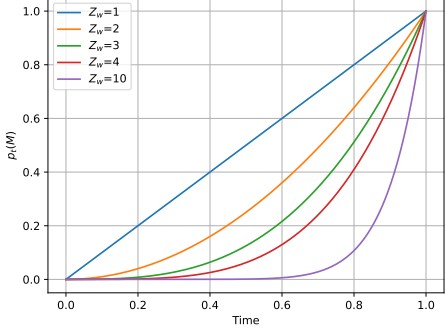

Figure 2: We plot the unmasking rates as a function of time under guidance. Faster unmasking ($Z_w > 1$) leads to worse numerical solvers, demonstrating an issue in the existing guidance mechanism.

Crucially, the partition function $Z_w$ appears in the exponent of the rate term, meaning that even small changes in $w$ can result in fast changes in the sampling rate. Figure 2 shows the percentage of tokens that remain masked as a function of time $p_t(M)$ for different values of $Z_w$. Applying guidance can significantly accelerate unmasking rates. While this can lead to faster generation, it may also introduce stiffness (Rathinam et al., 2003) and inefficiencies if not properly controlled.

### 3.2 Improved Guidance Mechanisms for Discrete Diffusion via Column Normalization

In order to alleviate the *unintentional* fast unmasking rates, we propose a simple yet effective change to the guidance mechanism. We begin by isolating the source of the issue by rewriting by decomposing the rate matrix in (2) into a *jump rate* and a *jump distribution*:

$$\overline{R}_t(y,x) = r_{t,p}(x) \underbrace{\left( \frac{1}{r_{t,p}(x)} \frac{p_t(y)}{p_t(x)} R_t(x,y) \right)}_{p_t(y|x)}, \qquad r_{t,p}(x) = \sum_{y \neq x} \overline{R}_t(y,x). \tag{9}$$

Under this decomposition, $r_{t,p}(x)$ governs *how often* jumps occur from state $x$, while $p_t(y \mid x)$ determines *where* the process jumps. Using this decomposition the guided rate matrix in (5) becomes:

$$\overline{R}_t^{(w)}(y,x) = \underbrace{r_{t,p}^w(x) r_{t,p}^{1-w}(x) \mathcal{Z}_w}_{\text{rate}} \quad \underbrace{\mathcal{Z}_w^{-1} p_t^w(y|x) q^{1-w}(y|x)}_{\text{distribution}} \tag{10}$$

Where $\mathcal{Z}_w = \sum_{y \neq x} p_t^w(y|x) q^{1-w}(y|x)$ is the normalizing constant of the guided jump measure. Crucially, (10) reveals that the $\mathcal{Z}_w$ does not merely affect the jump distribution, but instead *rescales the overall jump rate*. This is unintentional and leads to disproportionately fast transitions.

To make this effect explicit, we consider the masked diffusion setting and explicitly write the transition rates between a masked state $M$ a non-masked state:

**Lemma 3.1.** *The transition rates between a masked state and an unnormalized state are given by:*

$$\bar{R}_t^{(w)}(y,M) = R_t(x,y) \frac{e^{-\bar{\sigma}_t}}{1 - e^{-\bar{\sigma}_t}} Z_w p^{(w)}(y)$$

The appearance of $\mathcal{Z}_w$ as a multiplicative factor in the transition rate confirms the source of the observed pathology: guidance amplifies *how often* unmasking occurs, rather than only affecting *which* token is selected. Notably, when $w = 1$ (the purely conditional setting), $\mathcal{Z}_w = 1$ and the issue disappears, explaining why this effect is absent in standard conditional diffusion.

**Normalized Guidance.** The above analysis indicates that the normalizing constant $\mathcal{Z}_w$ should not influence the jump rate. To correct this, we explicitly decouple rate and distribution by normalizing the guided rate matrix column-wise:

$$\overline{R}_t^{(w)}(y,x) = \underbrace{r_{t,p}^w(x) r_{t,p}^{1-w}(x)}_{\text{rate}} \quad \underbrace{\mathcal{Z}_w^{-1} p_t^w(y|x) q^{1-w}(y|x)}_{\text{distribution}} \tag{11}$$

In the case of masked diffusion this normalization admits a very simple form:

$$\overline{R}_{\text{nor},t}^{(w)}(\hat{\mathbf{x}}, \mathbf{x}) = \frac{R_t(\mathbf{x}, \hat{\mathbf{x}}) e^{-\bar{\sigma}_t}}{1 - e^{-\bar{\sigma}_t}} \text{Softmax}(w \log p_0(\hat{\mathbf{x}}^i | \mathbf{x}^{\text{UM}}) + (1 - w) \log q_0(\hat{\mathbf{x}}^i | \mathbf{x}^{\text{UM}})). \tag{12}$$

The normalization introduced in (11) and (12) has the effect of smoothing the transport between the starting distribution and the data distribution. This simple change stabilizes the sampling process and allows for a cleaner theory. Notably, this can be done with a simple one line change to the code as presented in the pseudocode in 1. We further elaborate on the experimental benefits on Section 4.

### 3.3 Comparison of Guidance Mechanisms

We now clarify the distinctions between the various classifier-free guidance mechanisms. While some differences between our method and that of Nisonoff et al. (2024) were already discussed, we further highlight how our formulation also differs from the approach of Schiff et al. (2024). To better understand these differences, we begin by comparing the unlocking guidance mechanism of Nisonoff et al. (2024) with the simple guidance proposed by Schiff et al. (2024). For this analysis, we keep the guidance strength fixed throughout. Notice that: $p(x_s|x_t) = \exp\left( \int_s^t \overline{R}_\tau^{(w)} d\tau \right) p_t$. Therefore, if $p_t$ denotes the law of $x_t$, we can write the transition probabilities for each method:

$$p_{\text{unlocking}}(x_s|x_t) = \exp\left( \int_s^t \overline{R}_\tau^w(\cdot|c) \overline{R}_\tau^{1-w}(\cdot) d\tau \right) p_t,$$

$$p_{\text{simple}}(x_s|x_t) = Z_{\text{simple}}\Big( \exp\Big( \int_s^t \overline{R}_\tau(\cdot|c)\mathrm{d}\tau\Big)p_t\Big)^w \Big( \exp\Big( \int_s^t \overline{R}_\tau(\cdot)\mathrm{d}\tau\Big)p_t\Big)^{1-w}.$$

where $Z_{\text{simple}}$ is a normalizing constant. Now we look at the $w$-dependence inside the exponential. For $\log p_{\text{unlocking}}$, the $w$-dependence is *exponential* as it appears in the exponent of the rate matrices, while for $\log p_{\text{simple}}$, the $w$-dependence is *linear*. Therefore, the transitions induced by the unlocking guidance method get much more aggressive when $w$ increases. On the other hand, our normalization (depending on $w$) normalizes the columns so that it maintains the smoothness of the transition when $w$ increases. Therefore, our method approximates the convergence rates of the original process.

## 3.4 ANALYSIS OF GUIDANCE SCHEDULES IN 2D

Having addressed the existing issue we switch our focus to the analysis of guidance schedules in the case of two tokens. Although the analysis can be extended to higher-dimensions, the complexity of the problem grows exponentially with the dimension, leading to increasingly intricate expressions and reduced interpretability. This low-dimensional analysis already reveals the underlying mechanisms that define a good guidance schedule, and its impacts in high-dimensions are remarkable.

We start by stating our main theorem, in a simple to understand case that is used in practice. This simplification doesn't result in loss of generality, but significantly increases the interpretability of the results. We present a more general version in Theorem C.1.

**Corollary 3.1.** *Consider a time partition* $0 = t_0 < t_1 < t_2 < t_3 = T$ *with guidance* $w_i$ *in the interval* $[t_i, t_{i+1})$. *With* $\overline{\sigma} = -\log(1-\delta t)$ *and* $p_T(M, M) = 1$. *Then the sampled distribution follows the following formula:*

$$p_{t_0}(i,j) = \Big(\tfrac{t_3-t_2}{t_3}\Big)^2 p^{(w_2)}(i,j) + \Big(\tfrac{t_2-t_1}{t_3}\Big)^2 p^{(w_1)}(i,j) + \Big(\tfrac{t_1-t_0}{t_3}\Big)^2 p^{(w_0)}(i,j)$$

$$+ \tfrac{(t_3-t_2)(t_2-t_1)}{t_3^2}p^{(w_1,w_2)}(i,j) + \tfrac{(t_3-t_2)(t_1-t_0)}{t_3^2}p^{(w_0,w_2)}(i,j) \quad + \tfrac{(t_2-t_1)(t_1-t_0)}{t_3^2}p^{(w_0,w_1)}(i,j),$$

*where* $p^{(w,\gamma)}(i,j) = p^{(w)}(i,j|X_1 = i)p^{(\gamma)}(X_1 = i) + p^{(w)}(i,j|X_2 = j)p^{(\gamma)}(X_2 = j)$, *notice that this is not exactly a probability distribution as it is not normalized, but we will refer to it as one.*

This theorem states that guidance schedules induce an interpolation of different distributions, which depend only on the guidance strengths and that the portion assigned to each one depends on the time parameters. We analyze the role of each component separately.

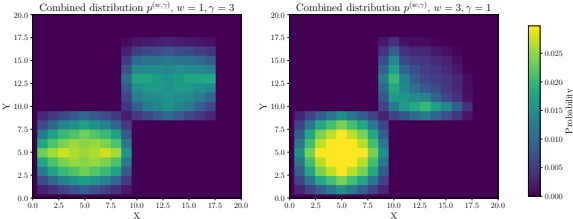

Figure 5: Notice that when $\omega < \gamma$ the combined distribution doesn't bias the leftmost mode, making this setting less efficient for guidance.

**The role of guidance weights**: We study a toy example in $2D$ with $4$ clusters, two of which intersect in the middle (see Appendix D for visualizations). Figure 3 shows that increasing $w$ leads to concentration of mass in one of the modes. Similarly, Figure 5 shows that $p^{(w,\gamma)}$ strongly resembles the tilted distribution of $w$. Practically, this means that the combined distribution will be more similar to the guidance applied at the end of the generation! Therefore, effective schedules have higher guidance at the final and middle phases of the generation while keeping early guidance small.

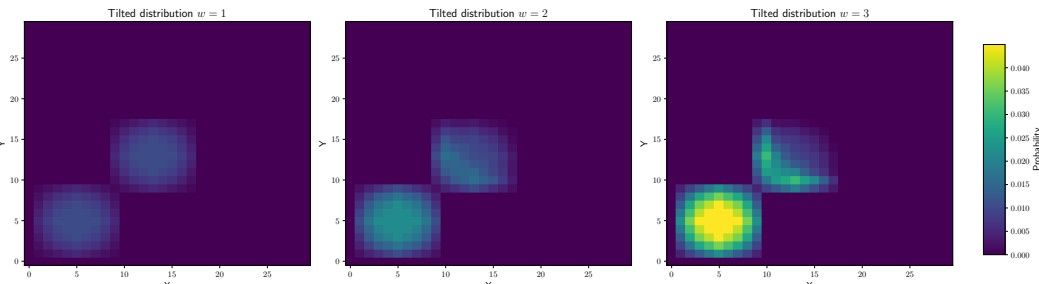

Figure 3: Tilted distributions for varying values of $w$. Large $w$ concentrates mass on one mode.

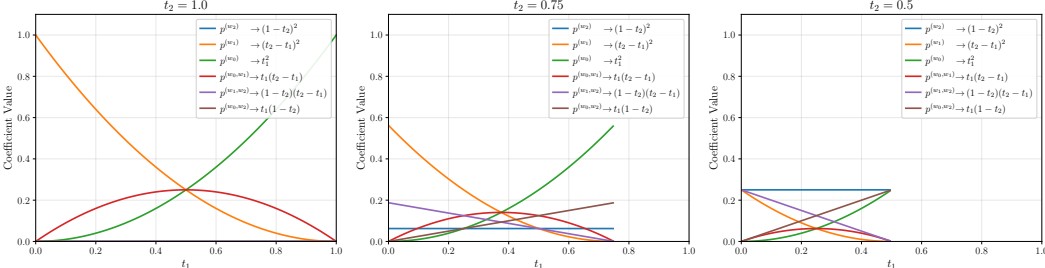

Figure 4: Evolution of the coefficients in Corollary 3.1 for different values of $t_2$, with $t_1 \leq t_2$. For moderate $t_2$, no single coefficient dominates, yielding a balanced target distribution.

**The role of the time parameters:** As observed in corollary 3.1, the time parameters set the proportion of each distribution that will contribute towards the final output. As observed in Figures 3,5, biasing just one of the distributions usually results in oversampling from a certain area. A good schedule is one that appropriately balances the contribution of each distribution.

We fix several values of $t_2$ and plot the coefficients as a function of $t_1$ in Figure 4. When $t_2 = 1$. we only have two intervals, and the curves change quickly; this implies that finding the right balance requires more careful tuning. On the other hand when $t_2 = .75$, many values of $t_1$ result in balanced combinations of all distributions, which ensures that we sample in a balanced way.

**Which schedules perform best?** Our theoretical analysis provides several insights into the design of effective guidance schedules. As discussed earlier, schedules that apply stronger guidance **during the middle and later stages** of the sampling process, while keeping early guidance small, tend to perform better. These selections seem to be the most critical, as they govern which distributions are mixed. Moreover, our theory predicts that using all **three intervals** (early, middle, and late) in the schedule facilitates **easier tuning** and yields more balanced output distributions. Based on these principles, we evaluate (according to our theory) various guidance schedules for discrete diffusion in Table 1, and we validate these predictions empirically in Section 4.2.

Table 1: Comparison of several guidance schedules.

|  | Low G. Beg | High G. Mid | High G. End | # Params Tune | Difficulty to Tune |
|---|---|---|---|---|---|
| **Constant** | ✗ | ✓ | ✓ | 1 | High |
| **Interval** | ✓ | ✓ | ✗ | 3 | Low |
| **Increasing** | ✓ | ✓ | ✓ | 1 | Low |
| **Decreasing** | ✗ | ✓ | ✗ | 1 | Low |

## 4 NUMERICAL RESULTS

In this section, we test whether low-dimensional theoretical insights extend to high-dimensional image and text domains. Section 4.1 studies the effect of normalization, while Section 4.2 examines different guidance schedules. Additional details and samples are provided in Appendix G.

### 4.1 EFFECT OF NORMALIZATION

Recall that our theory predicted that failing to normalize complicates the simulation, so normalization should improve results in practice, which we confirm below.

**Testing on Imagenet:** We assess MaskGIT on the ImageNet dataset (Deng et al., 2009) and evaluate FID on ImageNet-256 using 50K samples, following standard practices. For our method and for the Unlocking Guidance baseline (Nisonoff et al., 2024), we use the $\tau$-leaping sampler. For Simple Guidance (Schiff et al., 2024), we interpolate Euler transitions. For all methods, we use 50 steps. Figure 6a shows FID as a function of guidance strength using a constant schedule. Our experiments demonstrate that *failing to normalize can substantially degrade sample quality* as suggested by our theory.

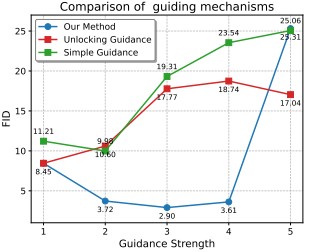 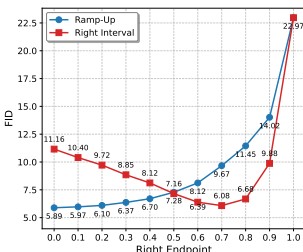 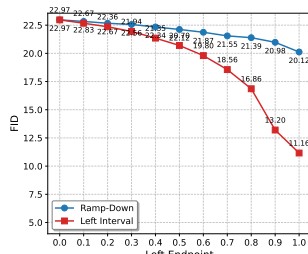

(a) Comparison of different guidance mechanisms.

(b) Right Interval vs Ramp-Up schedules.

(c) Left Interval vs Ramp-Down schedules

Figure 6: Evaluation of different guidance mechanisms and schedules on Imagenet

**The effect on diversity-quality tradeoff:** To assess the effect of guidance on the fidelity-diversity tradeoff, Fig. 7 shows the precision-recall curves. Relative to the no-guidance baseline ($w = 1$), both Unlocking Guidance and Simple Guidance exhibit reduced precision as guidance increases, indicating a tradeoff between fidelity and diversity. In contrast, our method improves precision while maintaining comparable recall for moderate guidance strengths, before all methods degrade at large guidance.

**Testing on text-to-image:** We evaluate our method on the GenEval benchmark (Ghosh et al., 2023) using Meissonic (Bai et al., 2024) as well as Show-O Xie et al. (2024) which is a mixed model leveraging discrete diffusion for image generation and auto-regressive for text generation. This benchmark provides a comprehensive measure of both prompt alignment and perceptual image quality. Figure 9 compares generations with and without normalization. Red regions indicate prompts where normalization improved the score. Overall, we observe consistent gains: *normalization enhances prompt adherence* and yields images that better match the target distribution.

**Testing on text generation:** To assess the effectiveness of normalization in the text generation domain, we evaluated using LLaDA-8B-Instruct (Nie et al., 2025) on the MATH-500 dataset, generating up to 256 tokens. We sample autoregressively in blocks of 32 tokens using a simple Euler sampler with 32 denoising steps per block, resulting in a total of 256 steps for the full generation.

Figure 8 presents the results of such an experiment. The results clearly show that *normalization consistently improves performance across all guidance strengths*. We note that the results are not directly comparable to those reported in the LLaDA paper; we use a simple Euler sampler without remasking to better isolate the effect of guidance and normalization in a simple setting.

**Empirical effect of normalization:** All our empirical findings demonstrate that normalization is a helpful step in improving the simulation of classifier-free guidance for discrete diffusion. This aligns with our low-dimensional theoretical analysis in Section 3.1, demonstrating that *low-dimensional studies can have a significant impact in high dimensions*.

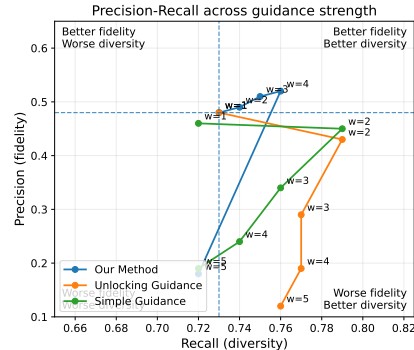

Figure 7: *Only our method simultaneously improves fidelity and diversity*. We plot the Precision/recall curves for different values of $w$. The dashed lines indicate the no-guidance baseline.

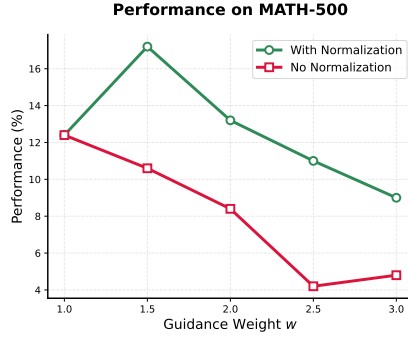

Figure 8: MATH-500 performance for LLada-8B-Instruct under a simple sampler without remasking to isolate the effect of the guidance mechanism. Normalization always yields better results.

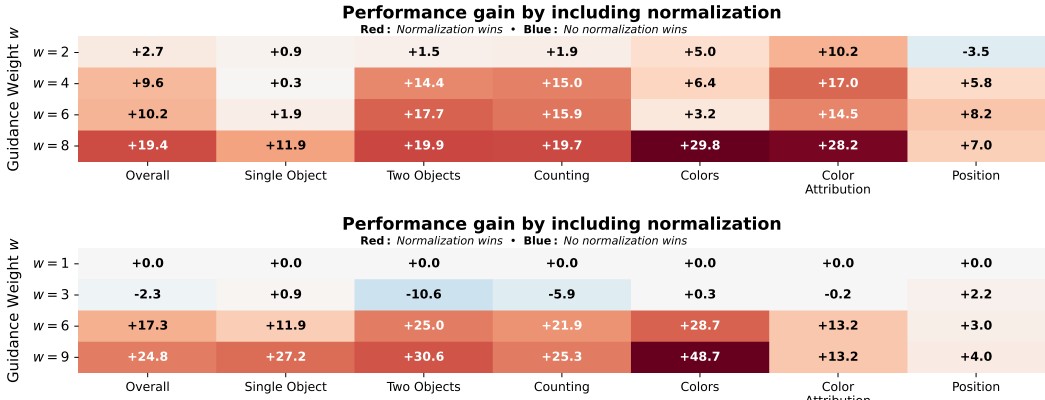

Figure 9: GenEval with and without normalization using Show-o (Top) and Meissonic (Bottom). Red denotes improved performance from normalization. Normalization improves prompt adherence and quality.

## 4.2 STUDY OF GUIDANCE SCHEDULES

Previously, our theory predicted that increasing schedules improve discrete diffusion while decreasing ones degrade generation. We test this theory on Imagenet-256 with 10K samples. For precise formulas for the schedules, see Table 2. When testing increasing schedules (Ramp-Up and Right Interval)

Table 2: Description of guidance schedules.

| Schedule | Formula $w(t)$ |
| --- | --- |
| **Left Interval** | $w \cdot \mathbf{1}_{[0,l]}(t)$ |
| **Right Interval** | $w \cdot \mathbf{1}_{[r,1]}(t)$ |
| **Ramp-Up** | $\min\left(w, w \cdot \frac{1-t}{1-r}\right)$ |
| **Ramp-Down** | $\min\left(w, w \cdot \frac{t}{\ell}\right)$ |

in 6b, we observe that both schedules can significantly improve the results. Furthermore, the Right Interval schedule exhibits a convex trend with respect to $r$, while the Ramp-Up schedule is monotone in $r$, and reaches a lower FID value, indicating that a gradual, linear increase in guidance outperforms abrupt alternatives. When testing the decreasing schedules (Left interval and Ramp-Down), we observe that they consistently damage the generation as seen in Figure 6c. Overall, *our experiments confirm our theory* that increasing schedules are most effective for masked diffusion.

## 5 CONCLUSIONS

In this work, we introduced a framework for analyzing guidance schedules in masked diffusion. Our analysis led to a novel approach for classifier-free guidance in the discrete setting. We validate the effectiveness of our method through experiments and show that guidance applied near $t = T$ is harmful to the generation quality while near $t = 0$ can improve the it. This insights enabled us to identify effective scheduling strategies. Our theoretical insights align closely with empirical observations, bridging the gap between theory and practice.

**Limitations and Future work.** While our framework provides a principled and tractable approach to CFG in discrete diffusion, our theoretical analysis is currently limited to masked diffusion in low-dimensional settings. Although the method is applicable to more complex real-world settings, our current theoretical study does not cover such regimes. Promising directions include extending the framework to other forms of discrete diffusion, such as uniform diffusion, scaling to higher dimensions, and analyzing the role of score estimation error in the guidance dynamics.

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

# A  NOTATION AND GENERAL RESULTS

## A.1  SPECIAL PROPERTIES OF MASKED DIFFUSION

We will use the following notations specific to masked diffusion. Let $\mathbf{x}_t = (\mathbf{x}_t^1, \ldots, \mathbf{x}_t^d)$ denote a random variable on $S$, and $M$ be the masked token. We will write $\mathbf{x}^{\text{UM}}$ for the set of elements such that $\mathbf{x}_t^i \neq M$, meaning the entries that are not the masked token. Additionally, we will denote $\overline{\sigma}_t = \int_0^t \sigma_s \mathrm{d}s$.

Masked diffusion has several appealing properties, one being the following shown by Ou et al. (2024):

**Lemma A.1.** *Along the dynamics* (1) *given by the masked rate matrix, if* $\mathbf{x}_t = (\mathbf{x}_t^1, \ldots, \mathbf{x}_t^d)$ *and* $\hat{\mathbf{x}}_t = (\mathbf{x}_t^1, \ldots, \hat{\mathbf{x}}_t^i, \ldots, \mathbf{x}_t^d)$ *in such a way that* $\hat{\mathbf{x}}_t^i \neq M$ *and* $\mathbf{x}_t^i = M$, *we have the following identity for the score*

$$\frac{p_t(\hat{\mathbf{x}}_t)}{p_t(\mathbf{x}_t)} = \frac{e^{-\overline{\sigma}_t}}{1 - e^{-\overline{\sigma}_t}} p_0(\hat{\mathbf{x}}_t^i | \mathbf{x}^{UM}).$$

This result is of great importance, as it tells us that it is possible to decompose the scores as a probability distribution independent of time multiplied by a time-dependent term.

# B  PROOFS IN 1D

We first prove a small lemma:

**Lemma B.1.** *Given a matrix of the form*

$$A = \begin{pmatrix} 0 & \cdots & 0 & v_1 \\ \vdots & \vdots & \vdots & \vdots \\ 0 & \cdots & 0 & v_n \end{pmatrix}$$

*If* $v_n \neq 0$, *then its matrix exponential is given by* $e^A = I + A \cdot \frac{e^{v_n} - 1}{v_n}$.

*Proof.* First notice that for $k > 0$ it holds that $A^k = v_n^{k-1} A$ then we can write:

$$\begin{aligned}
e^A &= I + A + \frac{1}{2!} A^2 + \frac{1}{3!} A^3 + \ldots \\
&= I + A + \frac{1}{2!} A v_n + \frac{1}{3!} A v_n^2 + \ldots \\
&= I + A(1 + \frac{1}{2!} v_n + \frac{1}{3!} v_n^2 + \ldots) \\
&= I + A(1 + \frac{1}{v_n}(\frac{1}{2!} v_n^2 + \frac{1}{3!} v_n^3 + \ldots) \\
&= I + A(1 + \frac{1}{v_n}(-1 - v_n + 1 + v_n + \frac{1}{2!} v_n^2 + \frac{1}{3!} v_n^3 + \ldots) \\
&= I + A(1 + \frac{1}{v_n}(-1 - v_n + e^{v_n}) \\
&= I + A \frac{e^{v_n} - 1}{v_n}
\end{aligned}$$

As we wanted. $\square$

We now state and prove the general version Theorem 3.1:

**Theorem B.1.** *Along the dynamics of equation* (7). *The distribution* $p_t$ *is given by:*

$$p_t = \left( A_1 \cdot \frac{1 - e^A}{A}, \quad \cdots \quad , A_{M-1} \cdot \frac{1 - e^A}{A}, \quad e^A \right)^\top.$$

*Where, for $i = 0, \ldots, M - 1$:*

$$A_i = \int_t^T \sigma_s \frac{e^{-\overline{\sigma}_s}}{1 - e^{-\overline{\sigma}_s}} Z_{w_s} \cdot p^{z,w_s}(i)ds, \qquad A = -\sum_{i=0}^{M-1} A_i = \int_t^T \sigma_s \frac{e^{-\overline{\sigma}_s}}{1 - e^{-\overline{\sigma}_s}} Z_{w_s}ds.$$

*Proof.* Recall that the rate matrix in the one-dimensional case is:

$$\overline{R}_t^{(w_t)} = \sigma_t \frac{e^{-\overline{\sigma}_t}}{1 - e^{-\overline{\sigma}_t}} Z_{w_t} \begin{pmatrix} 0 & 0 & \cdots & 0 & p^{(w_t)}(1) \\ 0 & 0 & \cdots & 0 & p^{(w_t)}(2) \\ \vdots & \vdots & \ddots & \vdots & \vdots \\ 0 & 0 & \cdots & 0 & p^{(w_t)}(M-1) \\ 0 & 0 & \cdots & 0 & -1 \end{pmatrix} \tag{13}$$

By direct integration we know that:

$$p_t = \exp\Big( \int_t^T \overline{R}_\tau^{(w_\tau)} d\tau \Big) p_T.$$

Therefore applying Lemma B.1 we get that (in vector notation):

$$p_t = p_T + p_T(M) \left( \frac{\int_t^T \sigma_s \frac{e^{-\overline{\sigma}_s}}{1 - e^{-\overline{\sigma}_s}} Z_{w_s} \cdot p^{z,w_s} ds \cdot \frac{1 - e^A}{A}}{e^A} \right),$$

with

$$A = -\sum_{i=0}^{M-1} A_i = \int_t^T \sigma_s \frac{e^{-\overline{\sigma}_s}}{1 - e^{-\overline{\sigma}_s}} Z_{w_s}ds.$$

The result is proved. $\qquad\square$

We can now use the previous theorem to compute the distribution under constant guidance:

**Corollary B.1.** *If we start with a distribution $p_t$ and keep guidance to be constant $w$. Then at time $s$ the distribution is given by:*

$$p_s(i) = p_t(i) + p_s(M)\Big( \frac{1 - e^{-\overline{\sigma}_t}}{1 - e^{-\overline{\sigma}_s}} - 1 \Big)^{Z_w} p^{(w)}(i)$$

*for $i \neq M$ and $p_s(M) = \Big( \frac{1 - e^{-\overline{\sigma}_t}}{1 - e^{-\overline{\sigma}_s}} - 1 \Big)^{Z_w} p_t(M)$*

*Proof.* The proof follows by keeping $w$ constant in the above theorem:

$$p_s = p_t + p_t(M) \left( \frac{\int_s^t \sigma_s \frac{e^{-\overline{\sigma}_s}}{1 - e^{-\overline{\sigma}_s}} Z_{w_s} \cdot p^{z,w_s} ds \cdot \frac{1 - e^A}{A}}{e^A} \right)$$

$$= p_t + p_t(M) \left( \frac{\int_t^T \sigma_s \frac{e^{-\overline{\sigma}_s}}{1 - e^{-\overline{\sigma}_s}} Z \cdot ds \cdot \frac{1 - e^A}{A} p^{(w)}}{e^A} \right)$$

$$= p_t + p_t(M) \left( \frac{(1 - e^A) p^{(w)}}{e^A} \right)$$

Substituting $A$ gives the desired result. $\qquad\square$

We can now chain the above argument to obtain a result for general piece-wise constant guidance schedules:

**Theorem B.2.** *Let $\delta = t_0 < t_1 < \cdots < t_k = T$ be a time partition and let $w_i$ the guidance strength on the interval $(t_i, t_{i+1}]$. Along the dynamics of equation (7), the sampled distribution $p_\delta$ is given by:*

$$p_\delta = p_T + \sum_{i=0}^{k-1} p_{t_{i+1}}(M) \cdot \left(1 - \left(\frac{1 - e^{-\bar{\sigma}_{t_i}}}{1 - e^{-\bar{\sigma}_{t_{i+1}}}}\right)^{Z_{w_i}}\right) p^{(w_i)}. \tag{14}$$

*Additionally, probability mass at $M$ at different time satisfies $p_{t_i}(M) = p_{t_{i+1}}(M)\left(\dfrac{1 - e^{-\bar{\sigma}_{t_i}}}{1 - e^{-\bar{\sigma}_{t_{i+1}}}}\right)^{Z_{w_i}}$ for all $i = 0, 1, \cdots, k-1$.*

**Lemma B.2.** *The transition rates between a masked state and an unnormalized state are given by:*

$$\bar{R}_t^{(w)}(y, M) = R_t(x, y)\frac{e^{-\bar{\sigma}_t}}{1 - e^{-\bar{\sigma}_t}}Z_w p^{(w)}(y)$$

*Proof.* Using Lemma A.1 we can write:

$$\begin{aligned}
\bar{R}_t^{(w)}(y, M) &= R_t(M, y) \cdot \left(\frac{p_t(x)}{p_t(M)}\right)^w \left(\frac{q_t(x)}{q_t(M)}\right)^{1-w} \\
&= R_t(M, y) \cdot \left(\frac{e^{-\bar{\sigma}_t}}{1 - e^{-\bar{\sigma}_t}}p_0(y)\right)^w \left(\frac{e^{-\bar{\sigma}_t}}{1 - e^{-\bar{\sigma}_t}}q_0(y)\right)^{1-w} \\
&= R_t(x, y)\frac{e^{-\bar{\sigma}_t}}{1 - e^{-\bar{\sigma}_t}}p_0^w(y)q_0^{1-w}(y) \\
&= R_t(x, y)\frac{e^{-\bar{\sigma}_t}}{1 - e^{-\bar{\sigma}_t}}Z_w p^{(w)}(y)
\end{aligned}$$

$\square$

The results for the normalized process are identical to the ones above, so we omit them for brevity.

## C    PROOFS IN 2D

We begin by writing a simple lemma that will come in handy later.

**Lemma C.1.** *Given a matrix of the form*

$$A = \begin{pmatrix} 0 & a & b & 0 \\ 0 & -1 & 0 & c \\ 0 & 0 & -1 & d \\ 0 & 0 & 0 & -2 \end{pmatrix}$$

*Then for any $\alpha \in \mathbb{R}$, it's matrix exponential is given by:*

$$\exp(\alpha A) = \begin{pmatrix} 1 & a(1 - e^{-\alpha}) & b(1 - e^{-\alpha}) & \frac{(ac+bd)(e^\alpha - 1)^2 e^{-2\alpha}}{2} \\ 0 & e^{-\alpha} & 0 & c\left(e^\alpha - 1\right)e^{-2\alpha} \\ 0 & 0 & e^{-\alpha} & d\left(e^\alpha - 1\right)e^{-2\alpha} \\ 0 & 0 & 0 & e^{-2\alpha} \end{pmatrix}$$

*Proof.* The proof of the above statement is easy by noticing that $A = PDP^{-1}$ with:

$$P = \begin{pmatrix} \frac{ac}{2} + \frac{bd}{2} & -a & -b & 1 \\ -c & 1 & 0 & 0 \\ -d & 0 & 1 & 0 \\ 1 & 0 & 0 & 0 \end{pmatrix}$$

$$D = \begin{pmatrix} -2 & 0 & 0 & 0 \\ 0 & -1 & 0 & 0 \\ 0 & 0 & -1 & 0 \\ 0 & 0 & 0 & 0 \end{pmatrix}$$

Then $\exp(\alpha A) = P\exp(\alpha D)P^{-1}$ and the result follows. $\square$

Now for the main proof we start by explicitly writing down the rate matrix in the case of two tokens. In this case the rate matrix will have the following structure:

$$\overline{R}^{(w)}_{\text{nor},t} = \frac{\sigma_t\,e^{-\bar{\sigma}_t}}{1 - e^{-\bar{\sigma}_t}} \begin{pmatrix} D_1 & \mathbf{0} & \dots & C_1 \\ \mathbf{0} & D_2 & \dots & C_2 \\ \vdots & \vdots & \ddots & \vdots \\ \mathbf{0} & \mathbf{0} & \mathbf{0} & L \end{pmatrix} := \frac{\sigma_t\,e^{-\bar{\sigma}_t}}{1 - e^{-\bar{\sigma}_t}} \overline{R}^{(w)}_{\text{nor}},$$

where each block is an $M \times M$ matrix given by the following formulas:

$$D_i = \begin{pmatrix} 0 & \dots & 0 & p^{(w)}(X_2 = 1 \mid X_1 = i) \\ 0 & \dots & 0 & p^{(w)}(X_2 = 2 \mid X_1 = i) \\ \vdots & \vdots & \vdots & \vdots \\ 0 & \dots & 0 & p^{(w)}(X_2 = M-1 \mid X_1 = i) \\ 0 & \dots & 0 & -1 \end{pmatrix}$$

$$C_i = \begin{pmatrix} p^{(w)}(X_1 = i \mid X_2 = 1) & 0 & \dots & & 0 \\ & \vdots & \ddots & \vdots & 0 \\ & 0 & \dots & p^{(w)}(X_1 = i \mid X_2 = M-1) & 0 \\ & 0 & \dots & 0 & p^{(w)}(X_1 = i) \end{pmatrix}$$

$$L = \begin{pmatrix} -1 & 0 & \dots & 0 & p^{(w)}(X_2 = 1) \\ 0 & -1 & \dots & 0 & p^{(w)}(X_2 = 2) \\ \vdots & \vdots & \ddots & \vdots & \vdots \\ 0 & \dots & 0 & -1 & p^{(w)}(X_2 = M-1) \\ 0 & \dots & 0 & 0 & -2 \end{pmatrix}$$

We can now state the main theorem:

**Theorem C.1.** *Given a starting distribution $p_t$ following the dynamics given by (7) the distribution at time $s$ is given by:*

$$p_s(i,j) = \begin{cases} p_t(i,j) + \left(1 - \dfrac{1 - e^{-\bar{\sigma}_s}}{1 - e^{-\bar{\sigma}_t}}\right)^2 p^{(w)}(i,j)\,p_t(M,M) \\[2mm] \quad + \left(1 - \dfrac{1 - e^{-\bar{\sigma}_s}}{1 - e^{-\bar{\sigma}_t}}\right) \Big[ p^{(w)}(X_2 = j \mid X_1 = i)\,p_t(i,M) \\[2mm] \quad + p^{(w)}(X_1 = i \mid X_2 = j)\,p_t(M,j) \Big] & \text{if } i, j \neq M \\[6mm] \left(\dfrac{1 - e^{-\bar{\sigma}_s}}{1 - e^{-\bar{\sigma}_t}}\right) p_t(i,M) \\[2mm] \quad + \left(\dfrac{1 - e^{-\bar{\sigma}_s}}{1 - e^{-\bar{\sigma}_t}}\right)^2 \left(\dfrac{1 - e^{-\bar{\sigma}_t}}{1 - e^{-\bar{\sigma}_s}} - 1\right) p^{(w)}(X_1 = i)p_t(M,M) & \text{if } i \neq M,\ j = M \\[6mm] \left(\dfrac{1 - e^{-\bar{\sigma}_s}}{1 - e^{-\bar{\sigma}_t}}\right) p_t(M,j) \\[2mm] \quad + \left(\dfrac{1 - e^{-\bar{\sigma}_s}}{1 - e^{-\bar{\sigma}_t}}\right)^2 \left(\dfrac{1 - e^{-\bar{\sigma}_t}}{1 - e^{-\bar{\sigma}_s}} - 1\right) p^{(w)}(X_2 = j)p_t(M,M) & \text{if } i = M,\ j \neq M \\[6mm] \left(\dfrac{1 - e^{-\bar{\sigma}_s}}{1 - e^{-\bar{\sigma}_t}}\right)^2 p_t(M,M) & \text{if } i = j = M \end{cases}$$

*Proof.* By direct integration we know that:

$$p_s = \exp\left(\int_s^t \frac{\sigma_\tau e^{-\bar{\sigma}_\tau}}{1 - e^{-\bar{\sigma}_\tau}}\,d\tau \overline{R}^{(w)}_{\text{nor}}\right) = \exp\left(\ln\left(\frac{1 - e^{-\bar{\sigma}_t}}{1 - e^{-\bar{\sigma}_s}}\right)\overline{R}^{(w)}_{\text{nor}}\right).$$

Due to the block structure of $\overline{R}_{\mathrm{nor}}^{(w)}$, it is enough to be able to compute the exponential of:

$$
\begin{pmatrix} D_i & C_i \\ \mathbf{0} & L \end{pmatrix} = \left[\begin{array}{cccc|cccc}
0 & \cdots & 0 & p^{(w)}(X_2=1\mid X_1=i) & p^{(w)}(X_1=i\mid X_2=1) & 0 & \cdots & 0 \\
0 & \cdots & 0 & p^{(w)}(X_2=2\mid X_1=i) & \vdots & & \ddots & \vdots & 0 \\
\vdots & \vdots & \vdots & \vdots & 0 & & \cdots & 0 \\
0 & \cdots & 0 & -1 & 0 & & \cdots & 0 & p^{(w)}(X_1=i) \\
0 & \cdots & 0 & 0 & -1 & 0 & \cdots & p^{(w)}(X_2=1) \\
0 & \cdots & 0 & 0 & 0 & -1 & \cdots & p^{(w)}(X_2=2) \\
\vdots & \vdots & \vdots & \vdots & \vdots & & \vdots & \ddots & \vdots \\
0 & \cdots & 0 & 0 & 0 & & \cdots & 0 & -2
\end{array}\right]
$$

where once again we can exploit the structured form of the matrix to simplify the calculation. It is clear that when computing products of this matrix, coordinates will only get affected by the smaller sublocks:

$$
\left(\begin{array}{cc|cc}
0 & p^{(w)}(X_2=j\mid X_1=i) & p^{(w)}(X_1=i\mid X_2=j) & 0 \\
0 & -1 & 0 & p^{(w)}(X_1=i) \\
0 & 0 & -1 & p^{(w)}(X_2=j) \\
0 & 0 & 0 & -2
\end{array}\right)
$$

This is not only clear from the structure, but it also reveals a true intuitive understanding. The probability mass at a given position can only be affected by those states that are reachable from the current state by masking or unmasking the entries. We can now use Lemma C.1 to find the exponential:

$$
\begin{pmatrix}
1 & p^{(w)}(X_2=j\mid X_1=i)(1-e^{-\alpha}) & p^{(w)}(X_1=i\mid X_2=j)(1-e^{-\alpha}) & p^{(w)}(i,j)(e^{\alpha}-1)^2 e^{-2\alpha} \\
0 & e^{-\alpha} & 0 & p^{(w)}(X_1=i)\,(e^{\alpha}-1)\,e^{-2\alpha} \\
0 & 0 & e^{-\alpha} & p^{(w)}(X_2=j)\,(e^{\alpha}-1)\,e^{-2\alpha} \\
0 & 0 & 0 & e^{-2\alpha}
\end{pmatrix}
$$

where $\alpha = \ln\left(\dfrac{1-e^{-\overline{\sigma}_t}}{1-e^{-\overline{\sigma}_s}}\right)$ and we used that $2p^{(w)}(i,j) = p^{(w)}(X_2=j\mid X_1=i)p^{(w)}(X_1=i) + p^{(w)}(X_1=i\mid X_2=j)p^{(w)}(X_2=j)$. Putting this together, we get that exponentiation each block we get:

$$
\left[\begin{array}{cccc|cccc}
1 & \cdots & 0 & p^{(w)}(X_2=1\mid X_1=i)(1-e^{-\alpha}) & p^{(w)}(X_1=i\mid X_2=1)(1-e^{-\alpha}) & \cdots & p^{(w)}(i,1)(e^{\alpha}-1)^2 e^{-2\alpha} \\
0 & \cdots & 0 & p^{(w)}(X_2=2\mid X_1=i)(1-e^{-\alpha}) & \vdots & \ddots & p^{(w)}(i,1)(e^{\alpha}-1)^2 e^{-2\alpha} \\
\vdots & \vdots & \vdots & \vdots & 0 & & \vdots \\
0 & \cdots & 0 & e^{-\alpha} & 0 & \cdots & p^{(w)}(X_1=i)(1-e^{\alpha})e^{-2\alpha} \\
0 & \cdots & 0 & 0 & e^{-\alpha} & \cdots & p^{(w)}(X_2=1)(1-e^{\alpha})e^{-2\alpha} \\
0 & \cdots & 0 & 0 & 0 & e^{-\alpha}\cdots & p^{(w)}(X_2=2)(1-e^{\alpha})e^{-2\alpha} \\
\vdots & \vdots & \vdots & \vdots & \vdots & \ddots & \vdots \\
0 & \cdots & 0 & 0 & 0 & \cdots & e^{-2\alpha}
\end{array}\right]
$$

With this, we have a full characterization of the matrix exponential. Therefore, we can simply write down the probability distribution by multiplying by $p_t$:

$$
p_s(i,j) = \begin{cases}
\begin{aligned}
&p_t(i,j) + (1 - e^{-\alpha})^2 \, p^{(w)}(i,j) \, p_t(M,M) \\
&\quad + (1 - e^{-\alpha}) \Big[ p^{(w)}(X_2 = j \mid X_1 = i) \, p_t(i,M) \\
&\quad + p^{(w)}(X_1 = i \mid X_2 = j) \, p_t(M,j) \Big]
\end{aligned} & \text{if } i,j \neq M \\[2em]
\begin{aligned}
&e^{-\alpha} p_t(i,M) \\
&\quad + e^{-2\alpha}(e^{\alpha} - 1) \, p^{(w)}(X_1 = i) \, p_t(M,M)
\end{aligned} & \text{if } i \neq M, \ j = M \\[2em]
\begin{aligned}
&e^{-\alpha} p_t(M,j) \\
&\quad + e^{-2\alpha}(e^{\alpha} - 1) \, p^{(w)}(X_2 = j) \, p_t(M,M)
\end{aligned} & \text{if } i = M, \ j \neq M \\[2em]
e^{-2\alpha} p_t(M,M) & \text{if } i = j = M
\end{cases}
$$

We can now replace $\alpha = \ln\left( \dfrac{1 - e^{-\overline{\sigma}_t}}{1 - e^{-\overline{\sigma}_s}} \right)$ into the formula above to obtain the result.

$\square$

**Corollary C.1.** *Given a starting distribution $p_t$ following the dynamics given by* (7) *with $\overline{\sigma}_t = -\log(1 - \delta t)$ the distribution at time $s$ is given by:*

$$
p_s(i,j) = \begin{cases}
\begin{aligned}
&p_t(i,j) + \left( \frac{t-s}{t} \right)^2 p^{(w)}(i,j) \, p_t(M,M) \\
&\quad + \left( \frac{t-s}{t} \right) \Big[ p^{(w)}(X_2 = j \mid X_1 = i) \, p_t(i,M) \\
&\quad + p^{(w)}(X_1 = i \mid X_2 = j) \, p_t(M,j) \Big]
\end{aligned} & \text{if } i,j \neq M \\[2em]
\frac{s}{t} \cdot p_t(i,M) + \left( \frac{s}{t} \right)^2 \left( \frac{t-s}{s} \right) p^{(w)}(X_1 = i) p_t(M,M) & \text{if } i \neq M, \ j = M \\[1.5em]
\frac{s}{t} \cdot p_t(M,j) + \left( \frac{s}{t} \right)^2 \left( \frac{t-s}{s} \right) p^{(w)}(X_2 = j) p_t(M,M) & \text{if } i = M, \ j \neq M \\[1.5em]
\left( \frac{s}{t} \right)^2 p_t(M,M) & \text{if } i = j = M
\end{cases}
$$

*Proof.* Notice that under this schedule we have that:

$$
\frac{1 - e^{-\overline{\sigma}_s}}{1 - e^{-\overline{\sigma}_t}} = \frac{\delta s}{\delta t} = \frac{s}{t}
$$

Substituting this in gives the corollary above. $\square$

*Proof of Corollary 3.1.* We track the changes in the distribution in every time interval. This can be found by plugging in the result of the corollary above. Firstly, on the interval $T \to t_2$ we obtain:

$$
p_{t_2}(M,M) = \left( \frac{t_2}{T} \right)^2
$$

$$p_{t_2}(M, j) = \left(\frac{t_2}{T}\right)^2 \left(\frac{T - t_2}{t_2}\right) p^{(w_2)}(X_2 = j)$$

$$p_{t_2}(i, M) = \left(\frac{t_2}{T}\right)^2 \left(\frac{T - t_2}{t_2}\right) p^{(w_2)}(X_1 = i)$$

$$p_{t_2}(i, j) = \left(\frac{T - t_2}{T}\right)^2 p^{(w_2)}(i, j)$$

Then on the interval from $t_2 \to t_1$ we get:

$$p_{t_1}(M, M) = \left(\frac{t_1}{T}\right)^2$$

$$p_{t_1}(M, j) = \left(\frac{t_1}{t_2}\right) p_{t_2}(M, j) + \left(\frac{t_1}{t_2}\right)^2 \left(\frac{t_2 - t_1}{t_1}\right) p^{(w_1)}(X_2 = j) p_{t_2}(M, M)$$

$$p_{t_1}(i, M) = \left(\frac{t_1}{t_2}\right) p_{t_2}(i, M) + \left(\frac{t_1}{t_2}\right)^2 \left(\frac{t_2 - t_1}{t_1}\right) p^{(w_1)}(X_1 = i) p_{t_2}(M, M)$$

$$p_{t_1}(i, j) = p_{t_2}(i, j) + \left(\frac{t_2 - t_1}{t_2}\right)^2 p^{(w_1)}(i, j) p_{t_2}(M, M)$$

$$+ \left(\frac{t_2 - t_1}{t_2}\right) [p^{(w_1}(X_2 = j | X_1 = i) p_{t_2}(i, M) + p^{(w_1)}(X_1 = i | X_2 = j) p_{t_2}(M, j)]$$

Replacing the values for $p_{t_1}$ into this equation we get:

$$p_{t_1}(M, M) = \left(\frac{t_1}{T}\right)^2$$

$$p_{t_1}(M, j) = \left(\frac{t_1(T - t_2)}{T^2}\right) p^{(w_2)}(X_2 = j) + \left(\frac{t_1}{T}\right)^2 \left(\frac{t_2 - t_1}{t_1}\right) p^{(w_2)}(X_2 = j)$$

$$p_{t_1}(i, M) = \left(\frac{t_1(T - t_2)}{T^2}\right) p^{(w_2)}(X_1 = i) + \left(\frac{t_1}{T}\right)^2 \left(\frac{t_2 - t_1}{t_1}\right) p^{(w_2)}(X_1 = i)$$

$$p_{t_1}(i, j) = \left(\frac{T - t_2}{T}\right)^2 p^{(w_2)}(i, j) + \left(\frac{t_2 - t_1}{t_2}\right)^2 \left(\frac{t_2}{T}\right)^2 p^{(w_1)}(i, j)$$

$$+ \left(\frac{t_2 - t_1}{t_2}\right) \left(\frac{t_2}{T}\right)^2 \left(\frac{T - t_2}{t_2}\right) p^{(w_1)}(X_2 = j | X_1 = i) p^{(w_2)}(X_1 = i)$$

$$+ \left(\frac{t_2 - t_1}{t_2}\right) \left(\frac{t_2}{T}\right)^2 \left(\frac{T - t_2}{t_2}\right) p^{(w_1)}(X_1 = i | X_2 = j) p^{(w_2)}(X_2 = j)$$

$$= \left(\frac{T - t_2}{T}\right)^2 p^{(w_2)}(i, j) + \left(\frac{t_2 - t_1}{T}\right)^2 p^{(w_1)}(i, j) + \frac{(t_2 - t_1)(T - t_2)}{T^2} p^{(w_1, w_2)}$$

Finally, we can proceed with the final step from $t_1 \to t_0$. In this case, we have:

$$p_{t_0}(M, M) = 0$$
$$p_{t_0}(M, j) = 0$$
$$p_{t_0}(i, M) = 0$$

$$p_{t_0}(i, j) = p_{t_1}(i, j) + \left(\frac{t_1 - t_0}{t_1}\right)^2 p^{(w_0)}(i, j) p_{t_1}(M, M)$$

$$+ \left(\frac{t_1 - t_0}{t_1}\right) [p^{(w_0}(X_2 = j | X_1 = i) p_{t_1}(i, M) + p^{(w_0)}(X_1 = i | X_2 = j) p_{t_1}(M, j)]$$

Then substituting in the previous results:

$$p_{t_0}(i, j) = p_{t_1}(i, j) + \left(\frac{t_1 - t_0}{t_1}\right)^2 \left(\frac{t_1}{T}\right)^2 p^{(w_0)}(i, j)$$

$$+ \left( \frac{t_1 - t_0}{t_1} \right) \Big[ p^{(w_0)}(X_2 = j | X_1 = i)$$

$$\left( \frac{t_1(T - t_2)}{T^2} p^{(w_2)}(X_1 = i) + \left( \frac{t_1}{T} \right)^2 \left( \frac{t_2 - t_1}{t_1} \right) p^{(w_2)}(X_1 = i) \right)$$

$$+ p^{(w_0)}(X_1 = i | X_2 = j)$$

$$\left( \frac{t_1(T - t_2)}{T^2} p^{(w_2)}(X_2 = j) + \left( \frac{t_1}{T} \right)^2 \left( \frac{t_2 - t_1}{t_1} \right) p^{(w_2)}(X_2 = j) \right) \Big]$$

Grouping by coefficient we get:

$$p_{t_0}(i,j) = p_{t_1}(i,j) + \left( \frac{t_1 - t_0}{t_1} \right)^2 \left( \frac{t_1}{T} \right)^2 p^{(w_0)}(i,j)$$

$$+ \left( \frac{t_1 - t_0}{t_1} \right) \cdot$$

$$\Big[ \left( \frac{t_1(T - t_2)}{T^2} \right) [p^{(w_0)}(X_2 = j | X_1 = i) p^{(w_2)}(X_1 = i) + p^{(w_0)}(X_1 = i | X_2 = j) p^{(w_2)}(X_2 = j)]$$

$$+ \left( \frac{t_1}{T} \right)^2 \left( \frac{t_2 - t_1}{t_1} \right) [p^{(w_0)}(X_2 = j | X_1 = i) p^{(w_1)}(X_1 = i) + p^{(w_0)}(X_1 = i | X_2 = j) p^{(w_1)}(X_2 = j)] \Big]$$

$$= p_{t_1}(i,j) + \left( \frac{t_1 - t_0}{t_1} \right)^2 \left( \frac{t_1}{T} \right)^2 p^{(w_0)}(i,j)$$

$$+ \left( \frac{t_1 - t_0}{t_1} \right) \cdot \Big[ \frac{t_1(T - t_2)}{T^2} p^{(w_0,w_2)} + \left( \frac{t_1}{T} \right)^2 \left( \frac{t_2 - t_1}{t_1} \right) p^{(w_0,w_1)} \Big]$$

Simplifying and substituting the term of $p_{t_1}$ this becomes:

$$p_{t_0}(i,j) = \left( \frac{t_3 - t_2}{t_3} \right)^2 p^{(w_2)}(i,j) + \left( \frac{t_2 - t_1}{t_3} \right)^2 p^{(w_1)}(i,j) + \left( \frac{t_1 - t_0}{t_3} \right)^2 p^{(w_0)}(i,j)$$

$$+ \frac{(t_3 - t_2)(t_2 - t_1)}{t_3^2} p^{(w_1,w_2)}(i,j) + \frac{(t_3 - t_2)(t_1 - t_0)}{t_3^2} p^{(w_0,w_2)}(i,j)$$

$$+ \frac{(t_2 - t_1)(t_1 - t_0)}{t_3^2} p^{(w_0,w_1)}(i,j).$$

$\square$

## D    DETAILS ON TOY EXAMPLE

We now present the details of the toy example that we used to demonstrate our theoretical results. In figure 10 we present plots of each class and the full data distribution. Each cluster is defined via the following matrix

$$\begin{bmatrix} 0.1 & 0.2 & 0.3 & 0.4 & 0.5 & 0.4 & 0.3 & 0.2 & 0.1 \\ 0.2 & 0.4 & 0.6 & 0.7 & 0.8 & 0.7 & 0.6 & 0.4 & 0.2 \\ 0.3 & 0.6 & 0.8 & 0.9 & 1.0 & 0.9 & 0.8 & 0.6 & 0.3 \\ 0.4 & 0.7 & 0.9 & 1.0 & 1.0 & 1.0 & 0.9 & 0.7 & 0.4 \\ 0.5 & 0.8 & 1.0 & 1.0 & 1.0 & 1.0 & 1.0 & 0.8 & 0.5 \\ 0.4 & 0.7 & 0.9 & 1.0 & 1.0 & 1.0 & 0.9 & 0.7 & 0.4 \\ 0.3 & 0.6 & 0.8 & 0.9 & 1.0 & 0.9 & 0.8 & 0.6 & 0.3 \\ 0.2 & 0.4 & 0.6 & 0.7 & 0.8 & 0.7 & 0.6 & 0.4 & 0.2 \\ 0.1 & 0.2 & 0.3 & 0.4 & 0.5 & 0.4 & 0.3 & 0.2 & 0.1 \end{bmatrix}$$

Each class is equally weighted. A pseudocode for generating the above dataset is:

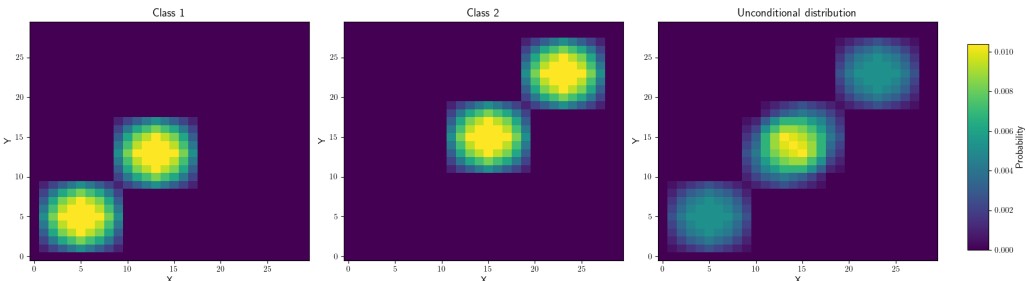

Figure 10: Definitions of the class and unconditional distributions for the toy problem.

```
height, width = 30, 30

matrix1 = torch.zeros((height, width))
matrix1[1:10, 1:10] = torch.tensor(cluster)
matrix1[9:18, 9:18] = torch.tensor(cluster)

matrix2 = torch.zeros((height, width))
matrix2[11:20, 11:20] = torch.tensor(cluster)
matrix2[19:28, 19:28] = torch.tensor(cluster)
```

Listing 3: Code to generate our toy dataset

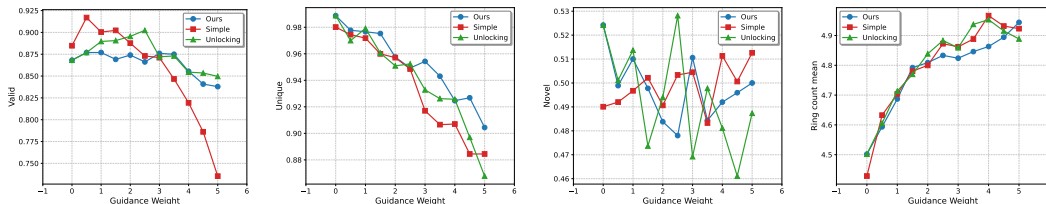

(a) Percentage of valid generated molecules

(b) Percentage of unique generated molecules

(c) Percentage of novel generated molecules

(d) Ring Count Mean as a function of guidance strength

Figure 11: We display the percentage of valid, unique, and novel molecules. We find that our method is the most robust to an increase in guidance strength.

## E   NORMALIZATION FOR GENERAL DIFFUSION PROCESSES

In this section we present a sample pseudocode for the general case in Listing E as well as some experiments using uniform diffusion in section E.1.

```python
def normalized_guidance_euler_transition(
        x, c, t, dt, w
    ):
    # Get scores
    log_score_c = get_score(x, t, cond=c)
    logs_core_u = get_score(x, t, cond=None)

    score_c = log_score_c.exp()
    score_u = log_score_u.exp()

    # Set diagonal terms to zero
    score_c.scatter_(-1, x[..., None], torch.zeros_like(score_c))
    score_u.scatter_(-1, x[..., None], torch.zeros_like(score_u))

    # Multiply matrix by edge to get the rates
    rate_c = edge * score_c
    rate_u = edge * score_u

    # Obtain the diagonal term
    total_rate_c = score_c.sum(dim=-1, keepdim=True)
    total_rate_u = score_u.sum(dim=-1, keepdim=True)

    # Get jump distributions
    jump_c = rate_c / total_rate_c
    jump_u = rate_u / total_rate_u

    # Get guided jump distribution
    jump_w = torch.softmax(w * jump_c + (1-w) * jump_u)
    total_jump_w = torch.exp(w * total_rate_c + (1-w) * total_rate_u)

    # Convert to the rate matrix
    rate_w = jump_w * total_jump_w
    rate_w.scatter_(-1, x[..., None], -total_jump_w)

    return sample(delta(x) + dt * sigma(t) * rate_w)
```

Listing 4: Our guidance in the general case using Euler transitions

### E.1   RESULTS ON QM9

We present results using our guidance mechanism in the context of uniform diffusion, applied to the QM9 small molecule dataset (Ruddigkeit et al., 2012; Ramakrishnan et al., 2014). QM9 is a dataset containing small organic molecules containing up to 9 heavy atoms. We train a conditional model on QM9 using uniform diffusion, based on the official implementation of Schiff et al. (2024), without modifying the architecture or hyperparameters. The model is conditioned on the number of rings in each molecule (ring count). Unlike ImageNet, evaluation on QM9 is more nuanced: we generate 1,024 samples and assess several metrics. First, generated molecules must satisfy chemical constraints to be considered valid. Second, a key goal of generative models is to produce novel molecules not found in the training data. Accordingly, we report both validity and novelty, along with the mean ring count (i.e., the conditioning signal), in Figure 11.

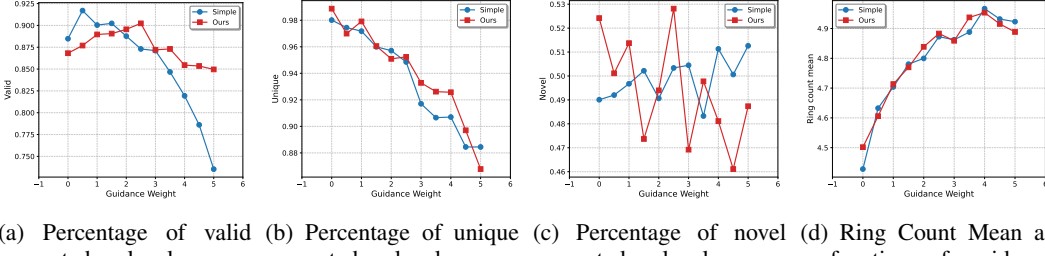

(a) Percentage of valid generated molecules

(b) Percentage of unique generated molecules

(c) Percentage of novel generated molecules

(d) Ring Count Mean as a function of guidance strength

Figure 12: We display the percentage of valid, unique, and novel molecules. We find that our method is the most robust to an increase in guidance strength.

We find that our method is the most robust to increases in guidance strength. However, in general, all methods perform comparably well across the full range of strengths. This suggests that normalization may have a less pronounced effect in the uniform diffusion setting. Due to the complexity of evaluating on QM9, further experiments on additional discrete datasets are needed to more conclusively determine the optimal guidance mechanism. We leave this for future work.

## F    EXTRA EXPERIMENTS ON QM9 FOR MASKED DIFFUSION

We present similar results using our guidance mechanism but in the context of masked diffusion, applied to the QM9 small molecule dataset (Ruddigkeit et al., 2012; Ramakrishnan et al., 2014). We train a conditional model on QM9 using uniform diffusion, based on the official implementation of Schiff et al. (2024), without modifying the architecture or hyperparameters. The model is conditioned on the number of rings in each molecule (ring count). We report statistics under the same setting as in Appendix E.

We plot the results in Figure 12, we find that our method is the most robust to increases in guidance strength and generally achieves better results across various guidance strengths. Suggesting that normalization is also a helpful technique on this domain.

# G  GENERATED SAMPLES TEXT TO IMAGE

To generate these samples we made use of a single node with $8$ NVIDIA A100 GPUs. We present samples to compare our method against other guidance methods as well as the detailed results of the GenEval benchmark in Table 3 and 4. The results demonstrate that normalization is key in order to improve the sample quality.

Table 3: Performance comparison across different guidance weights using Meissonic as a base model

| Metric | Ours | | | | Unlocking | | | |
|---|---|---|---|---|---|---|---|---|
| | $w=1$ | $w=3$ | $w=6$ | $w=9$ | $w=1$ | $w=3$ | $w=6$ | $w=9$ |
| Overall | 8.2 | 40.8 | 45.9 | 44.7 | 8.2 | 43.1 | 28.5 | 19.9 |
| Objects Single | 23.8 | 89.4 | 91.9 | 91.2 | 23.8 | 88.4 | 80.0 | 64.1 |
| Objects Two | 3.0 | 36.9 | 48.2 | 48.7 | 3.0 | 47.5 | 23.2 | 18.2 |
| Counting | 0.6 | 27.2 | 33.8 | 28.4 | 0.6 | 33.1 | 11.9 | 3.1 |
| Position | 20.7 | 72.3 | 77.4 | 77.7 | 20.7 | 72.3 | 48.7 | 29.0 |
| Color Attribution | 0.2 | 8.5 | 7.8 | 7.8 | 0.2 | 6.2 | 7.8 | 3.8 |
| Colors | 1.0 | 10.5 | 16.5 | 14.5 | 1.0 | 10.8 | 2.8 | 1.0 |

Table 4: Performance comparison across different guidance weights using Show-o as a base model

| Metric | Ours | | | | Unlocking | | | |
|---|---|---|---|---|---|---|---|---|
| | $w=2$ | $w=4$ | $w=6$ | $w=8$ | $w=2$ | $w=4$ | $w=6$ | $w=8$ |
| Overall | 56.42 | 62.46 | 63.13 | 63.39 | 53.73 | 52.84 | 52.89 | 43.96 |
| Objects Single | 96.88 | 98.75 | 99.06 | 98.75 | 95.94 | 98.44 | 97.19 | 86.88 |
| Objects Two | 65.66 | 75.76 | 78.28 | 80.05 | 64.14 | 61.36 | 60.61 | 60.10 |
| Counting | 41.56 | 50.00 | 50.94 | 50.94 | 39.69 | 35.00 | 35.00 | 31.25 |
| Position | 78.19 | 81.38 | 79.79 | 81.12 | 73.14 | 75.00 | 76.60 | 51.33 |
| Color Attribution | 22.75 | 28.5 | 30.25 | 27.25 | 26.25 | 22.75 | 22.00 | 20.25 |
| Colors | 33.5 | 41.5 | 40.5 | 42.25 | 23.25 | 24.50 | 26.00 | 14.00 |

## G.1 GENERATED SAMPLES FROM MEISSONIC

We now present some samples from our method:

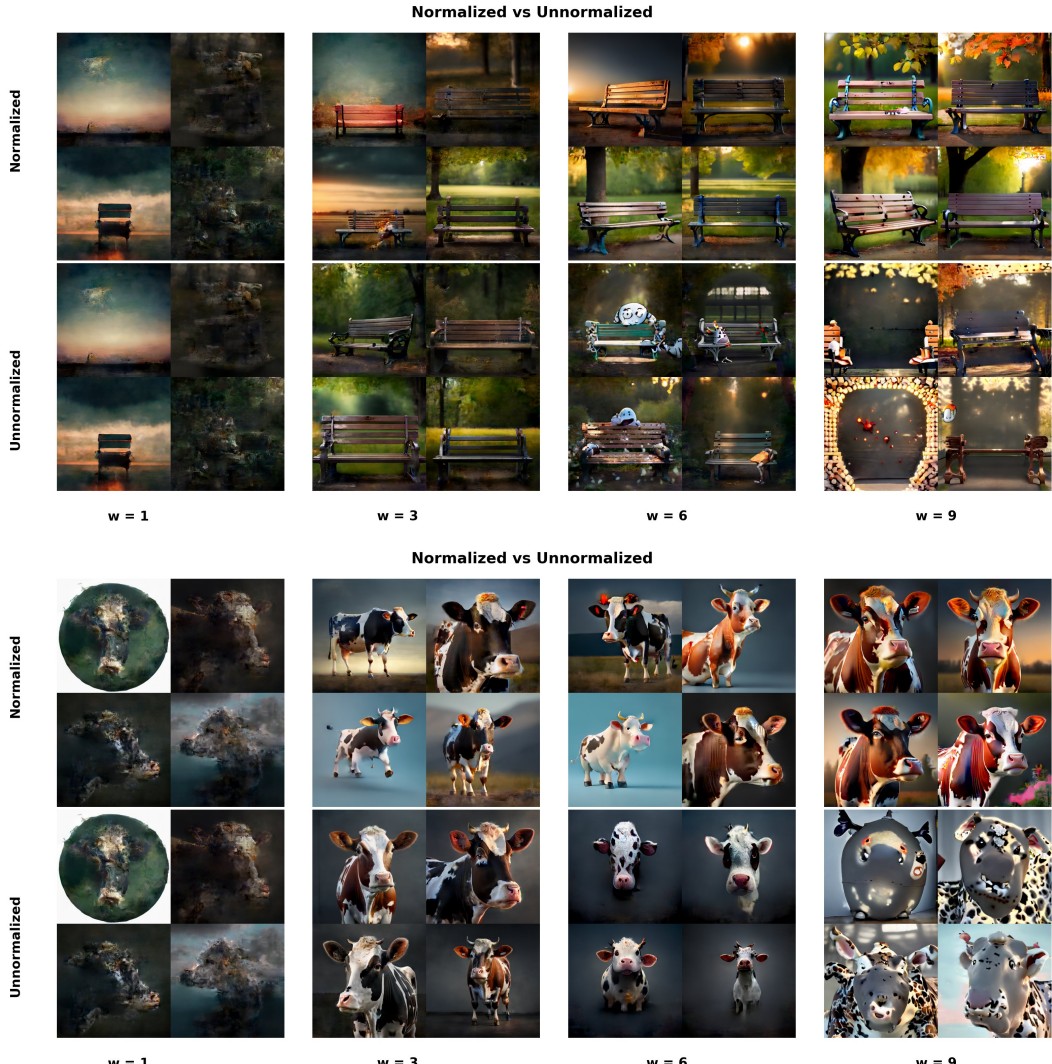

Figure 13: Comparison of samples generated by different guidance methods across various seeds and configurations. Using the prompts: A photo of a bench (top), A photo of a cow (bottom).

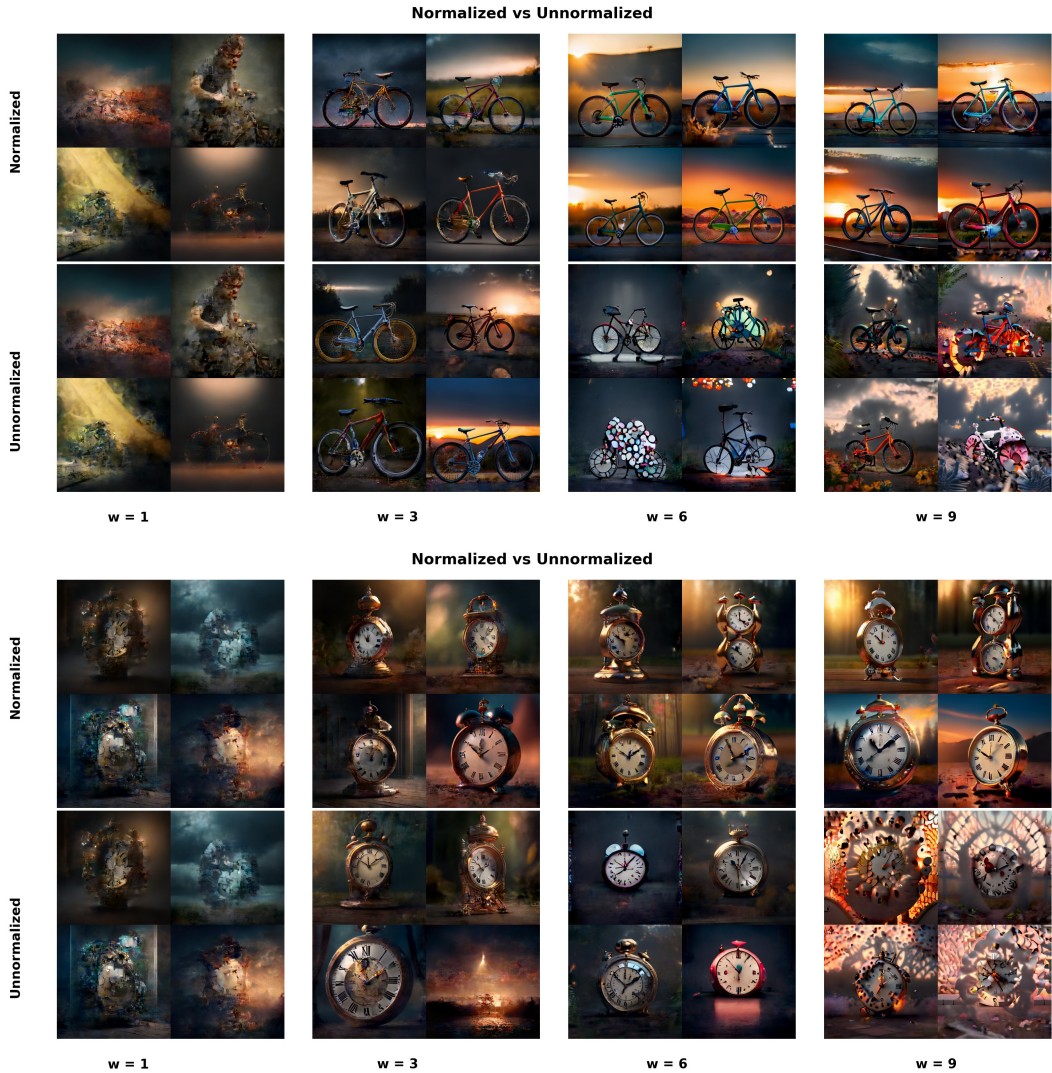

Figure 14: Comparison of samples generated by different guidance methods across various seeds and configurations. Using the prompts: A photo of a bike (top), A photo of a clock (bottom).

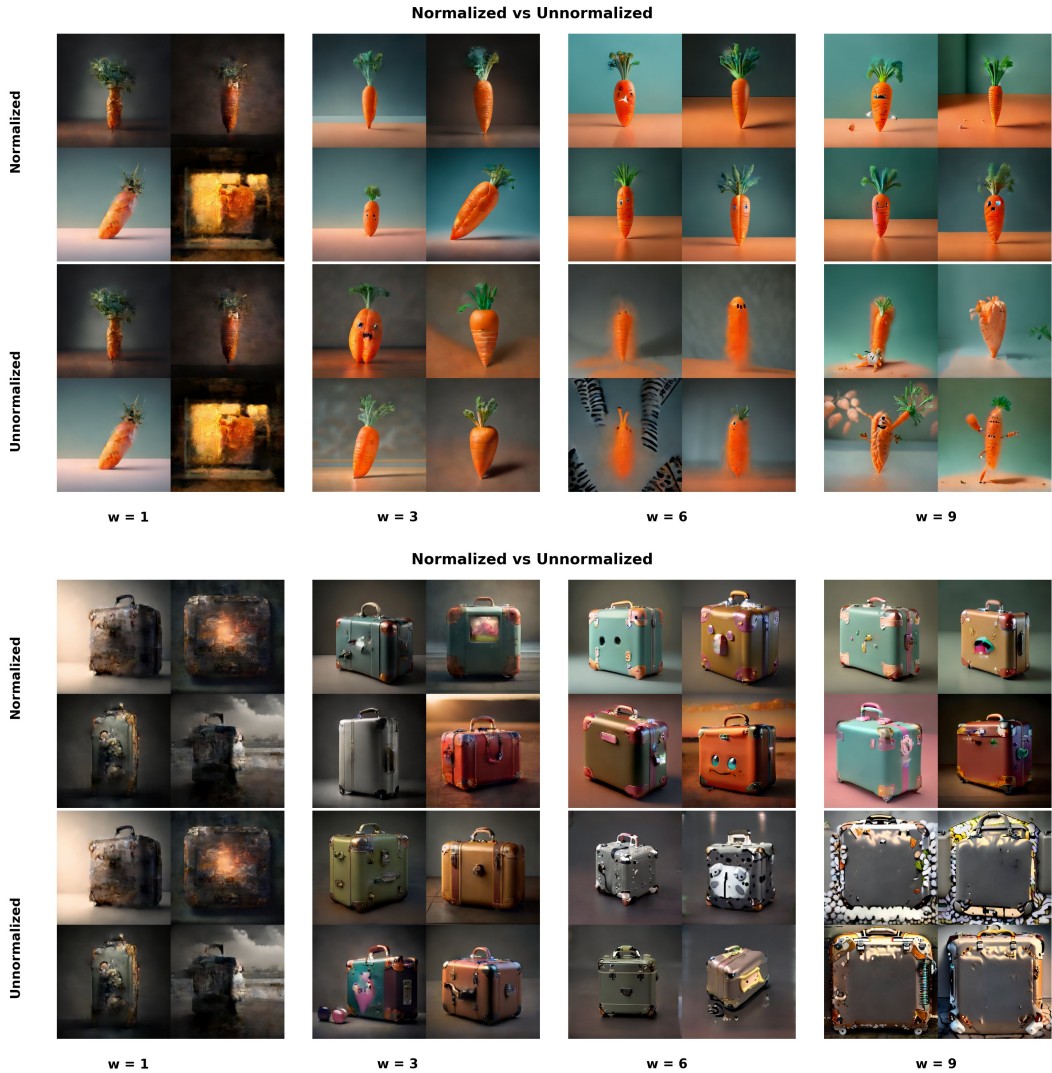

Figure 15: Comparison of samples generated by different guidance methods across various seeds and configurations. Using the prompts: A photo of a carrot (top), A photo of a suitcase (bottom).

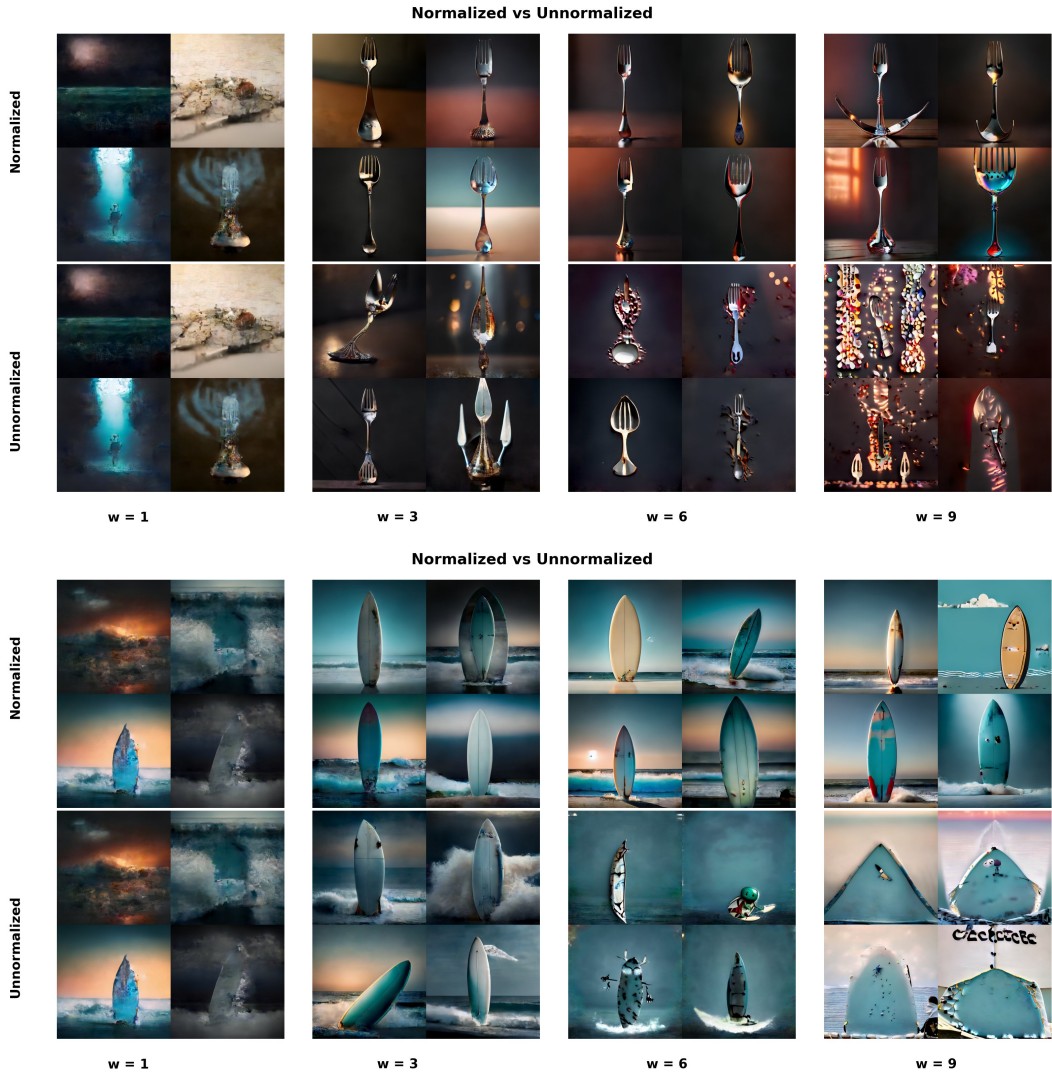

Figure 16: Comparison of samples generated by different guidance methods across various seeds and configurations. Using the prompts: A photo of a fork (top), A photo of a surfboard (bottom).

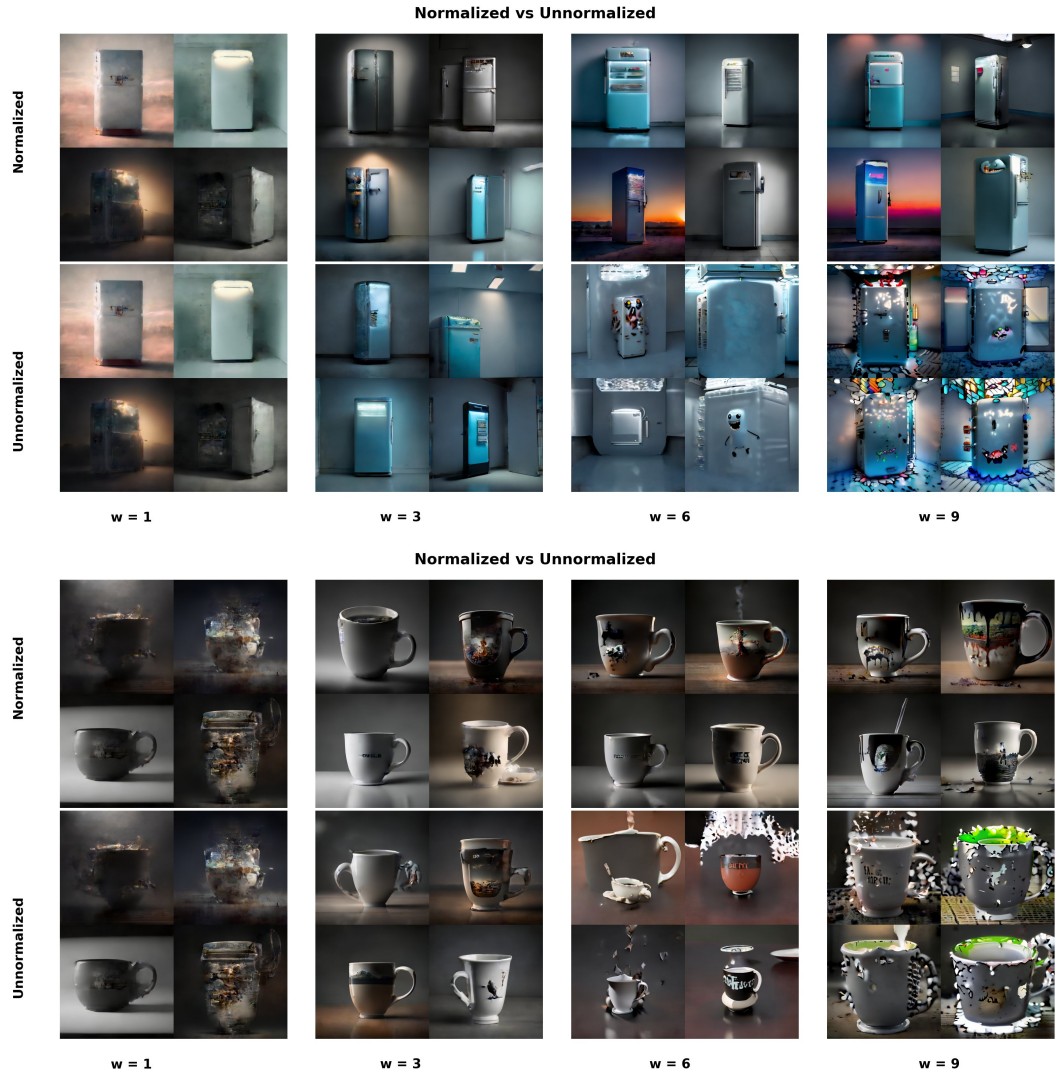

Figure 17: Comparison of samples generated by different guidance methods across various seeds and configurations. Using the prompt: A photo of a refrigerator (top), A photo of a cup (bottom)

## G.2 GENERATED SAMPLES FROM SHOW-O

We now present some samples from our method using Show-o Xie et al. (2024):

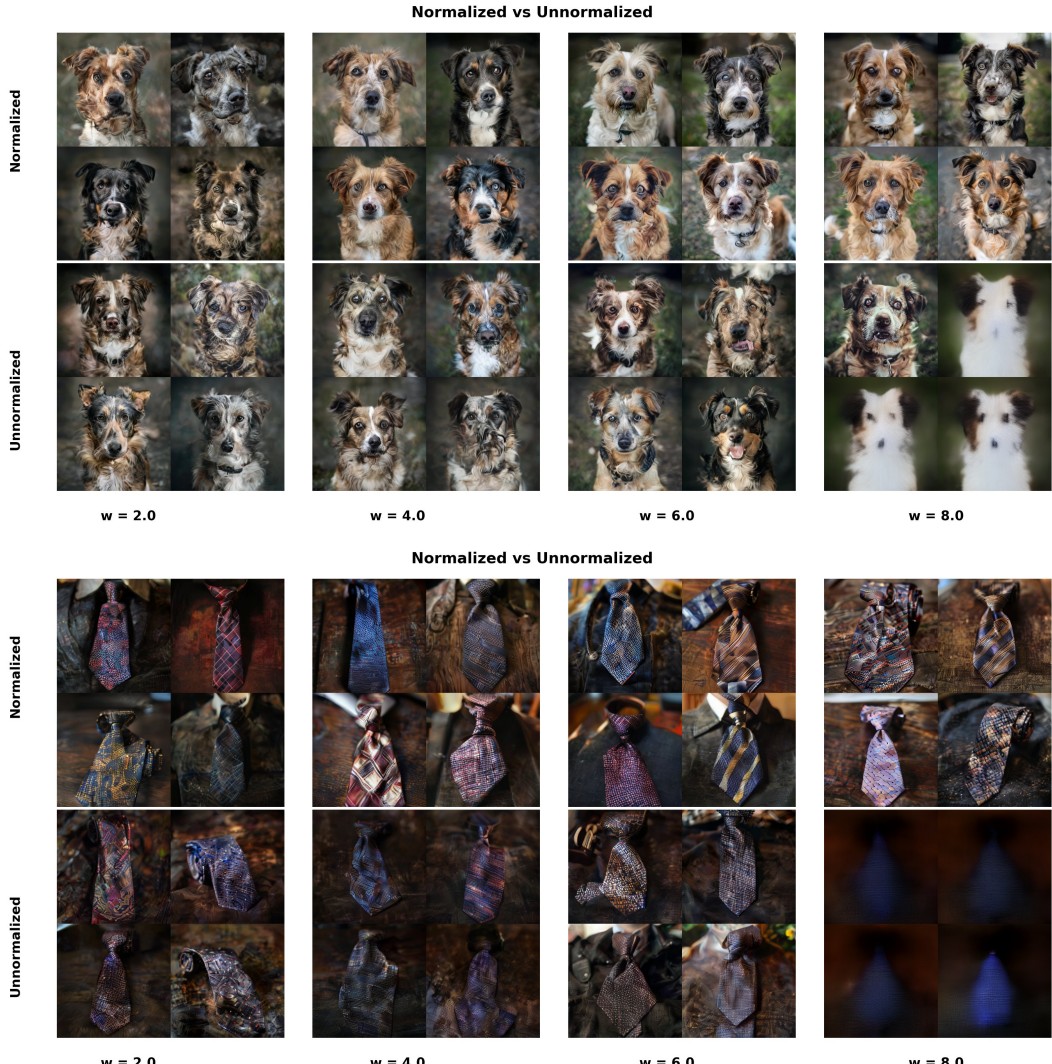

Figure 18: Comparison of samples generated by different guidance methods across various seeds and configurations. Using the prompts: A photo of a dog (top), A photo of a tie (bottom).

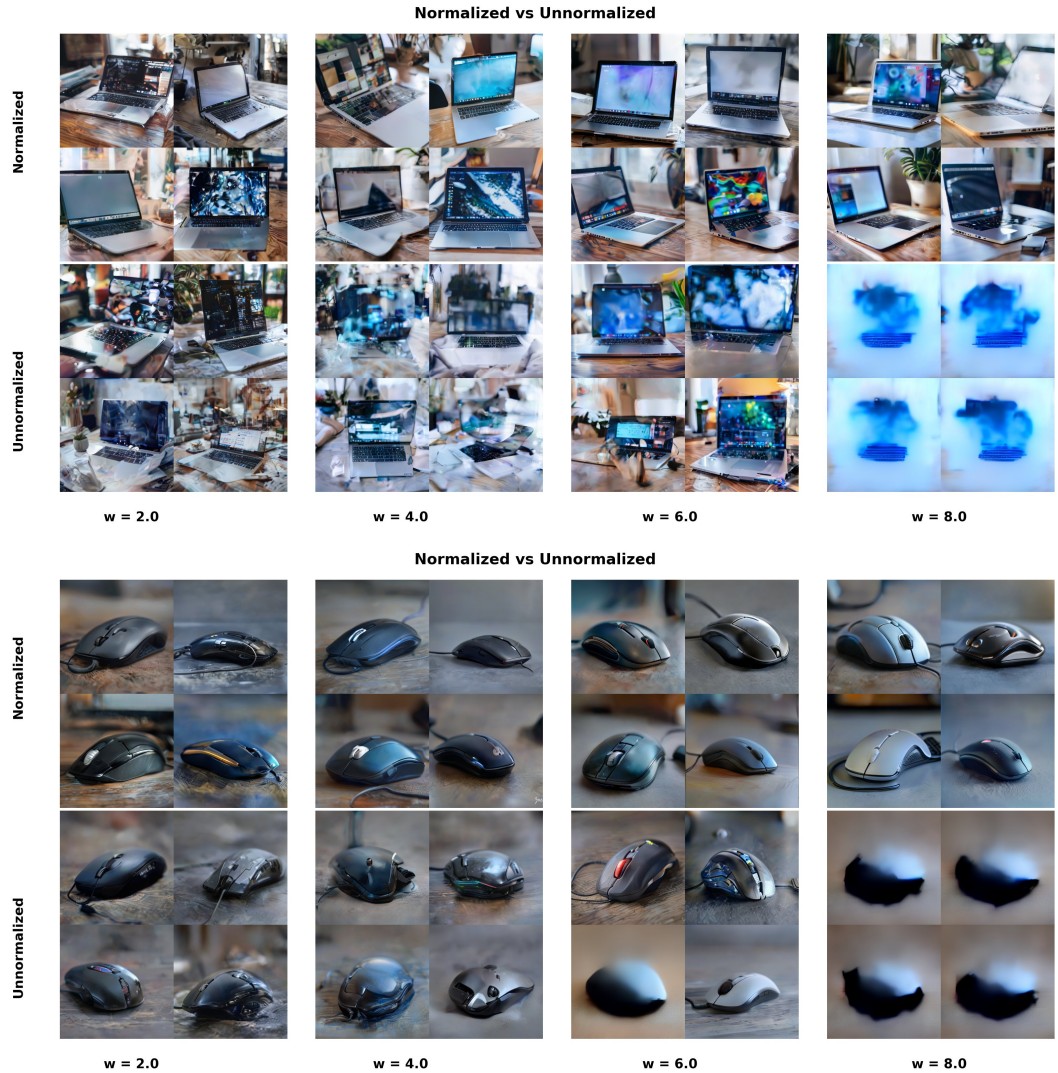

Figure 19: Comparison of samples generated by different guidance methods across various seeds and configurations. Using the prompts: A photo of a laptop (top), A photo of a computer mouse (bottom).

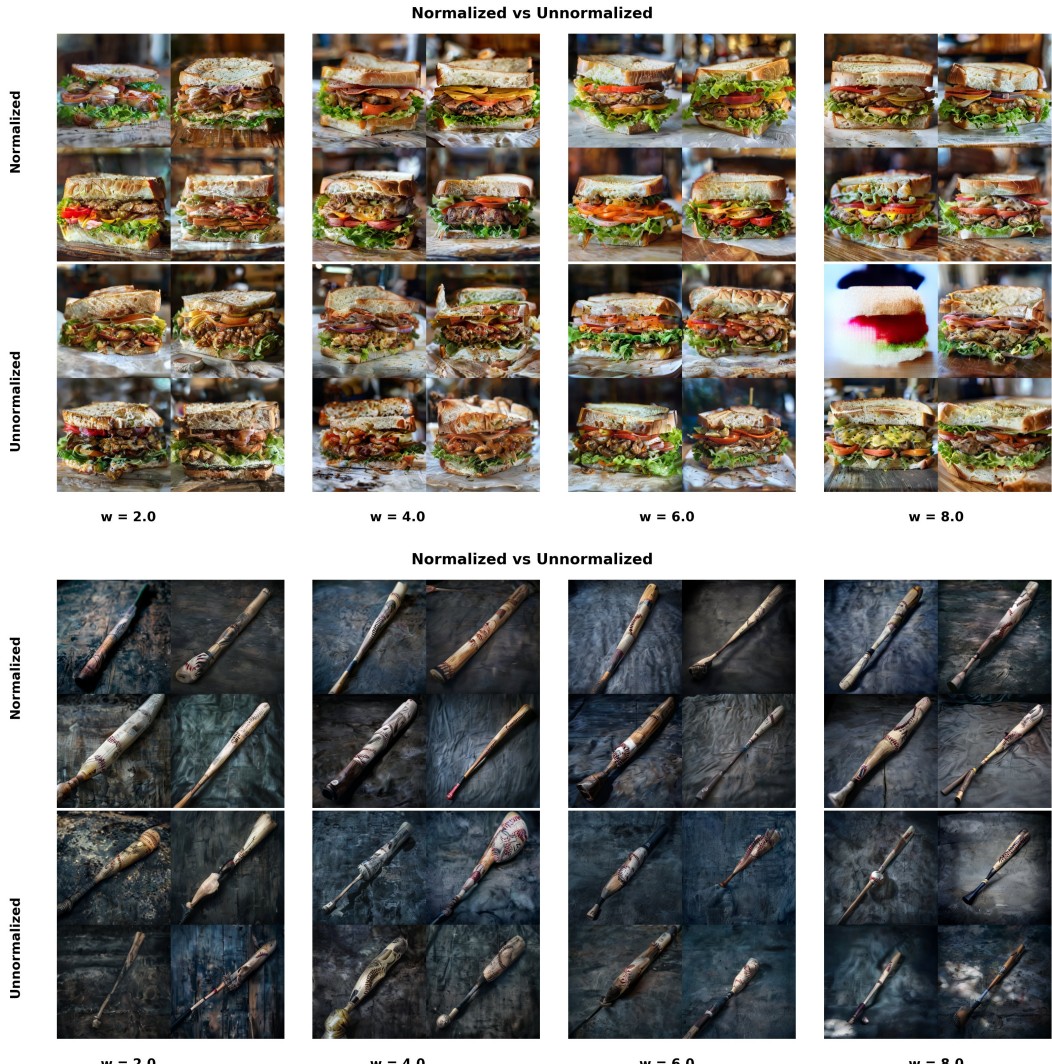

Figure 20: Comparison of samples generated by different guidance methods across various seeds and configurations. Using the prompts: A photo of a sandwich (top), A photo of a baseball bat (bottom).

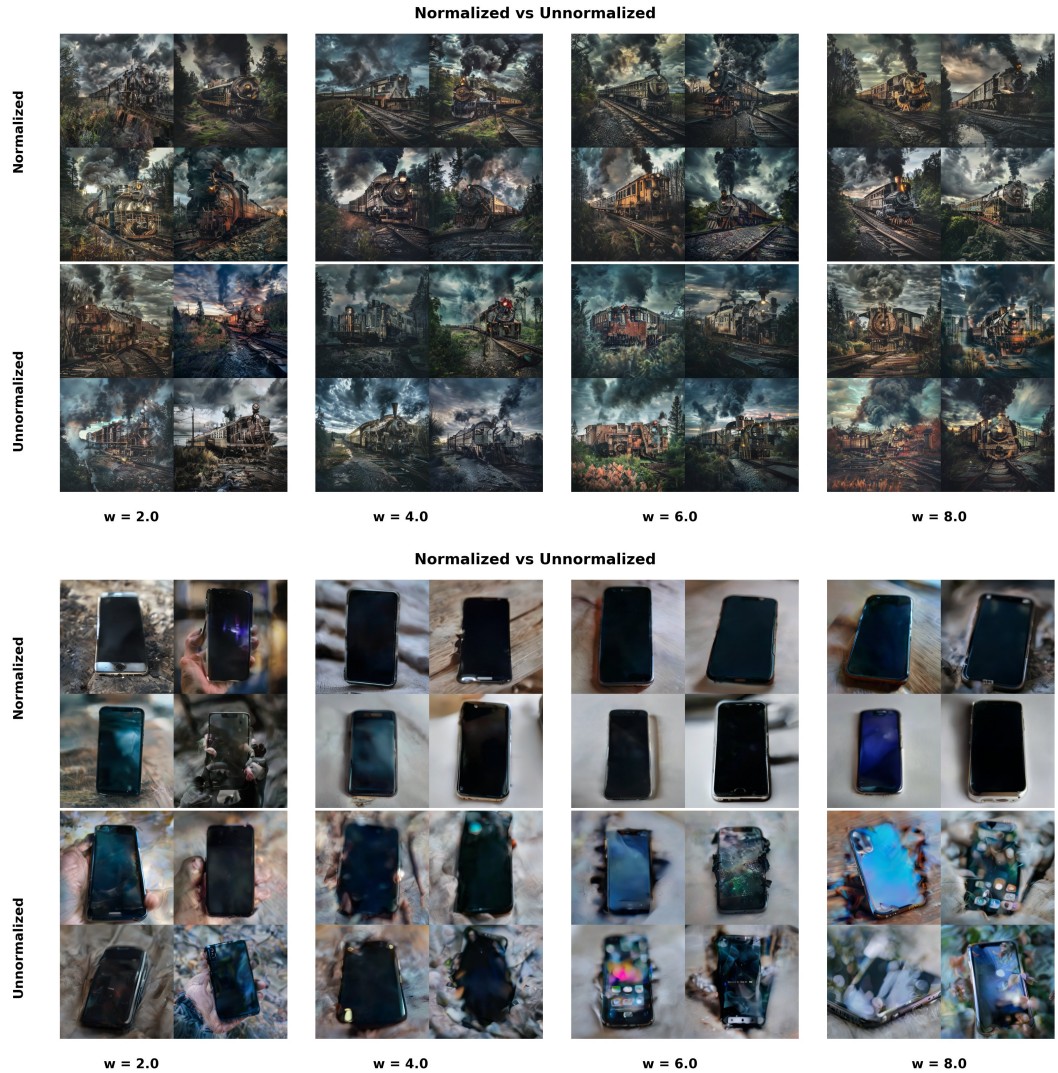

Figure 21: Comparison of samples generated by different guidance methods across various seeds and configurations. Using the prompts: A photo of a train (top), A photo of a cell phone (bottom).

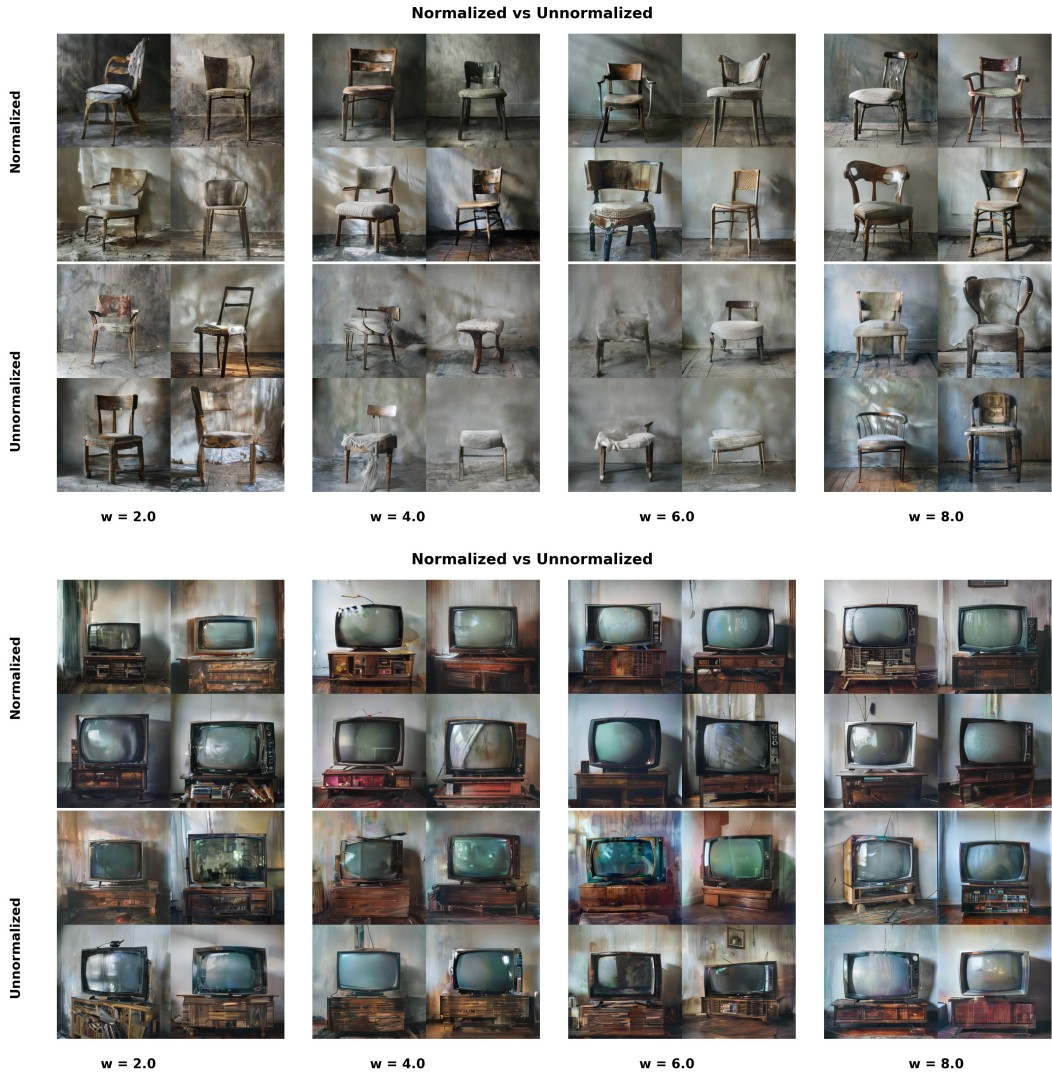

Figure 22: Comparison of samples generated by different guidance methods across various seeds and configurations. Using the prompt: A photo of a chair (top), A photo of a tv (bottom)

# H GENERATED SAMPLES IMAGENET

## H.1 GUIDANCE STRENGTH $w = 2$

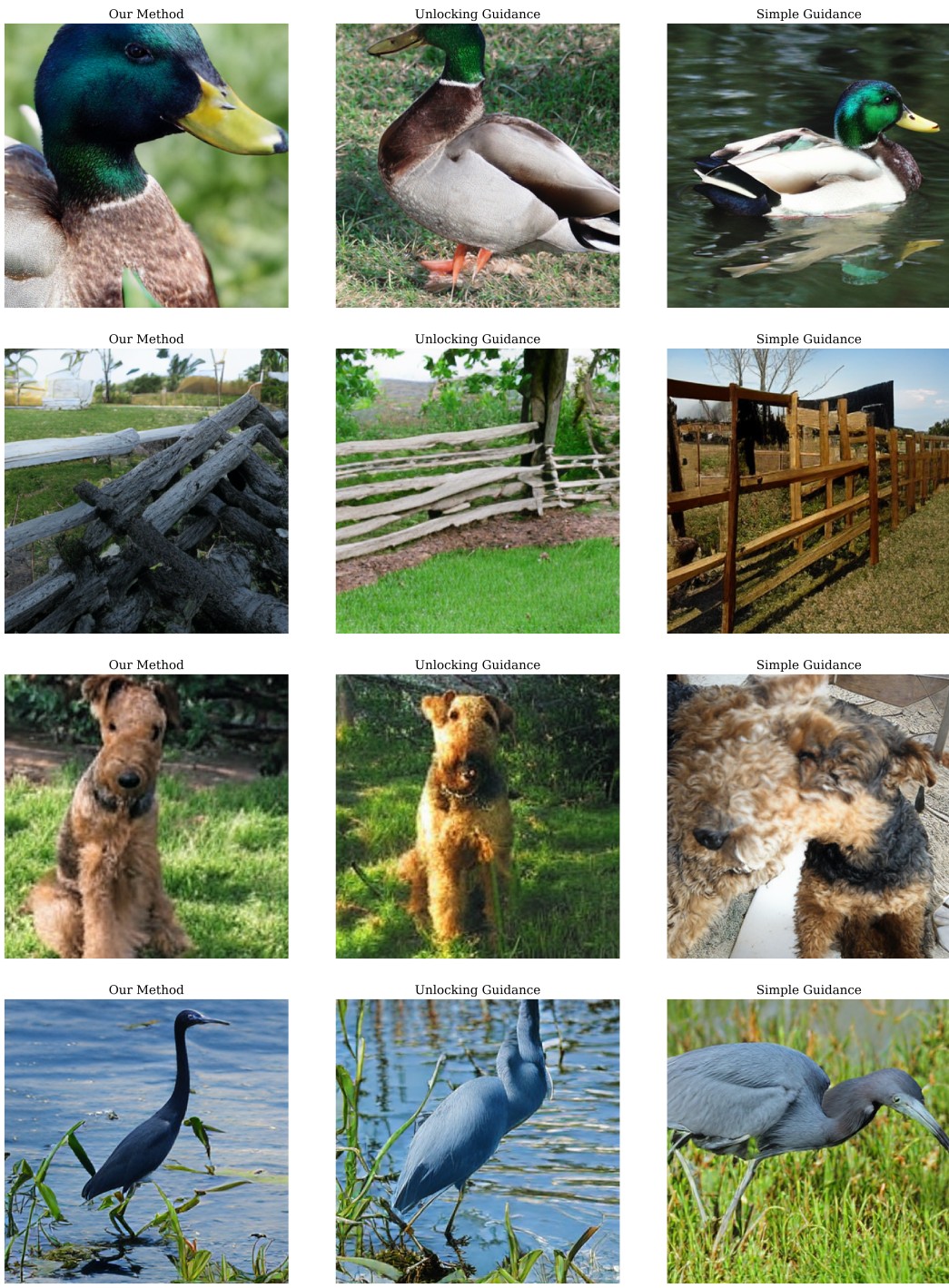

Figure 23: Comparison of samples generated by different guidance methods across various seeds or configurations.

## H.2 GUIDANCE STRENGTH $w = 3$

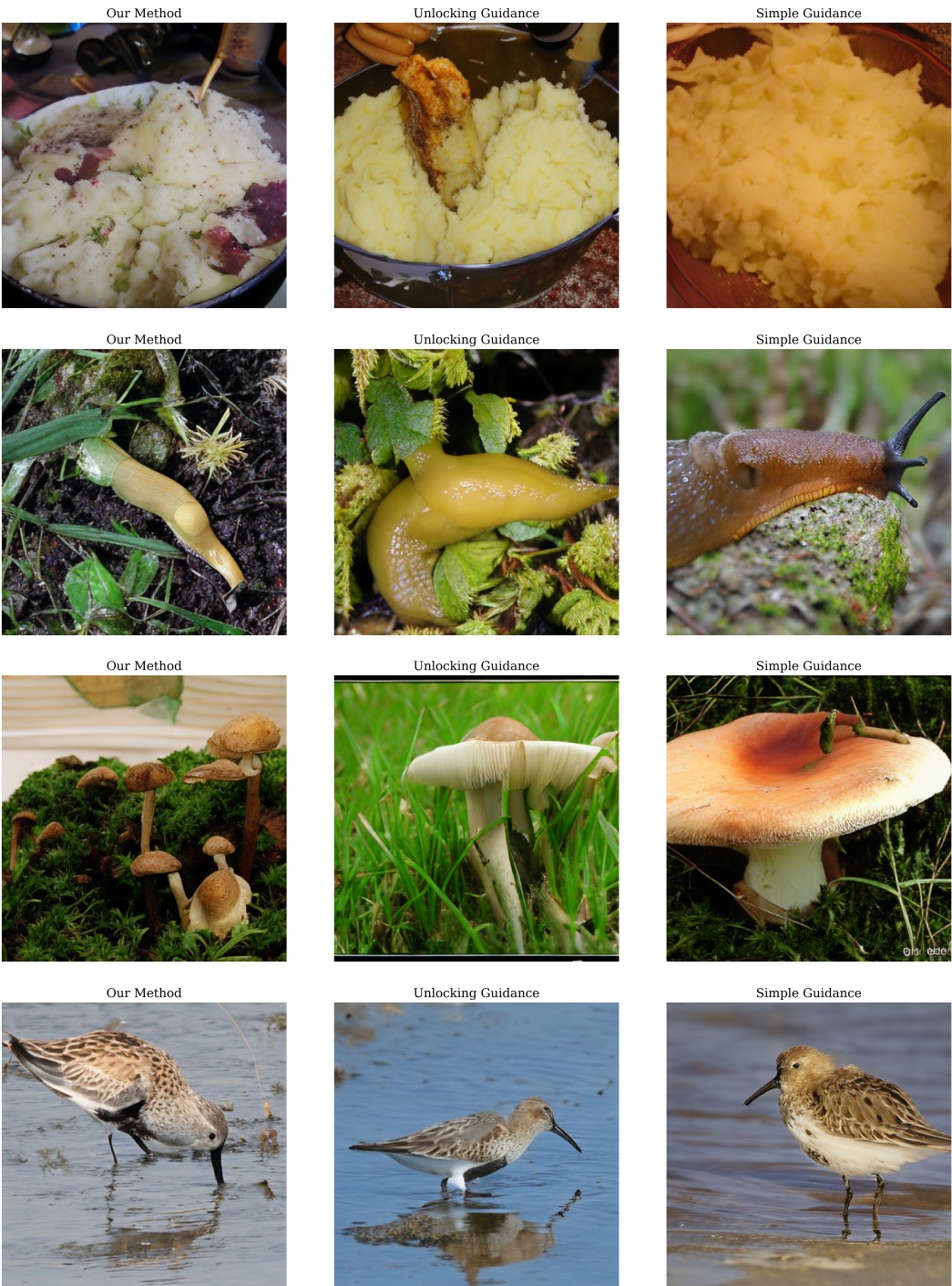

Figure 24: Comparison of samples generated by different guidance methods across various seeds or configurations.

## H.3 GUIDANCE STRENGTH $w = 4$

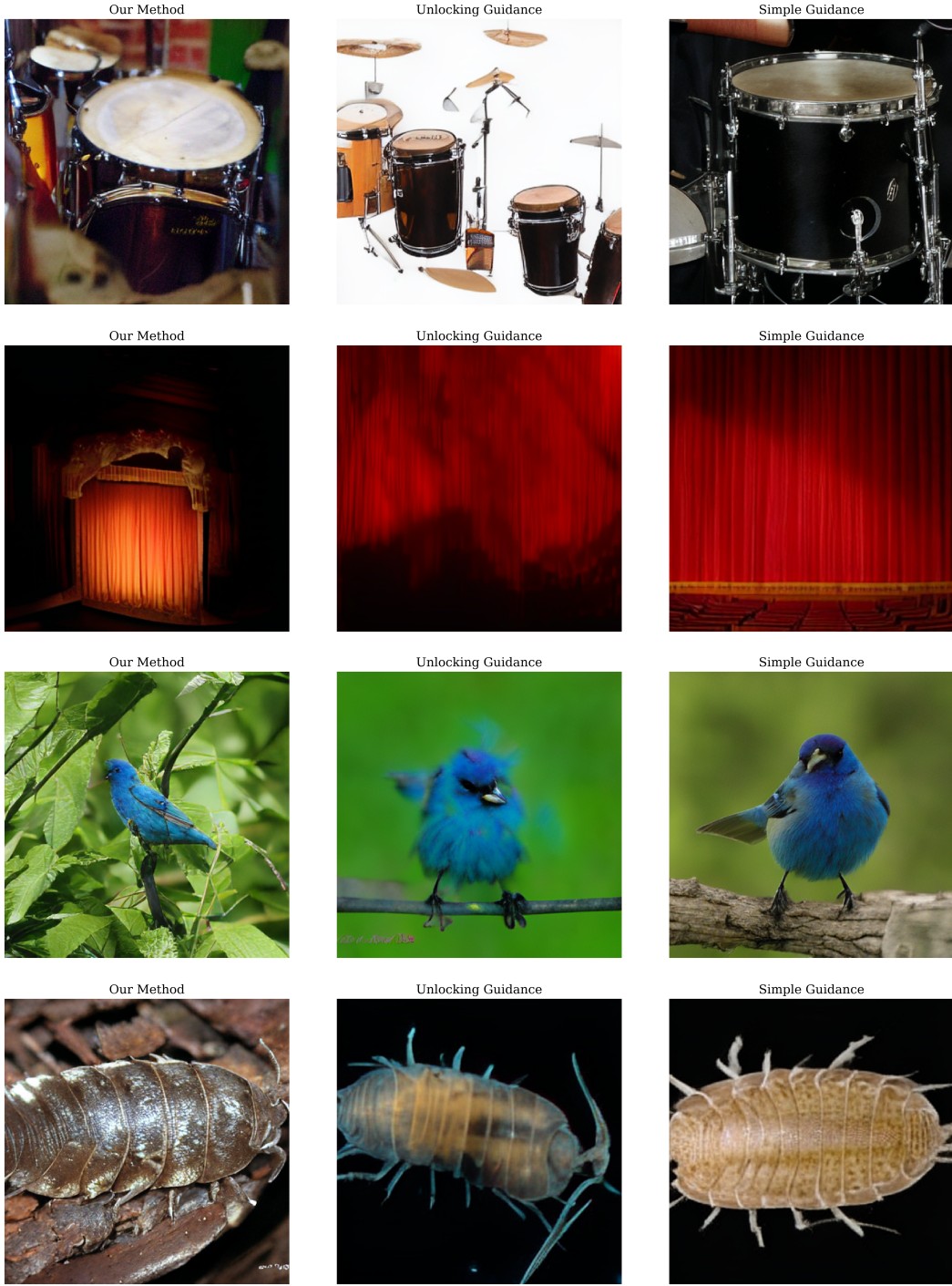

Figure 25: Comparison of samples generated by different guidance methods across various seeds or configurations.

## H.4 Guidance strength $w = 5$

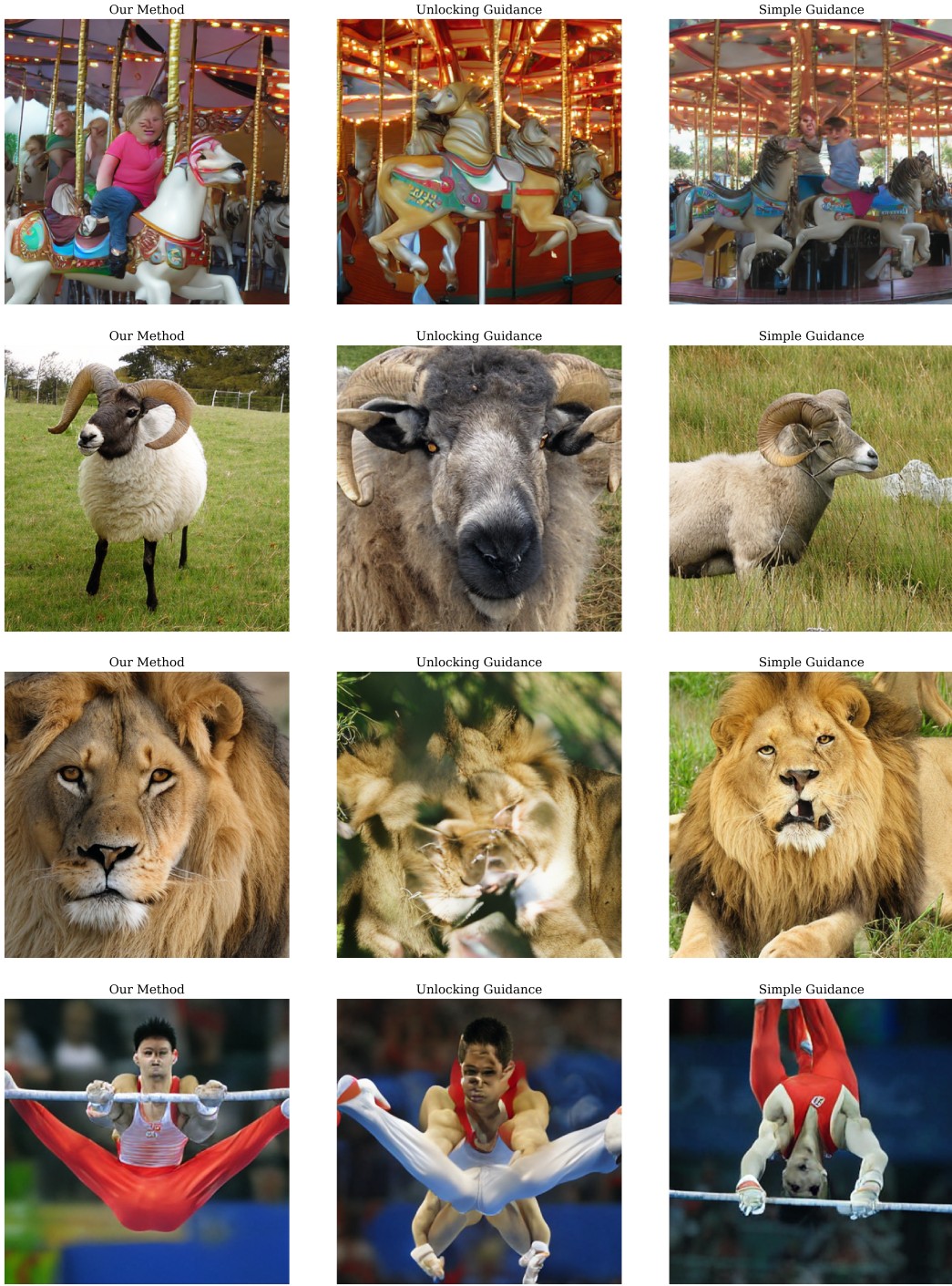

Figure 26: Comparison of samples generated by different guidance methods across various seeds or configurations.

## STATEMENT ON THE USE OF LARGE LANGUAGE MODELS

This work made use of large language models to assist with proofreading and improving the clarity of the writing. All research ideas, theoretical development, and experiments were carried out solely by the authors. When used for coding, it was solely used for plotting purposes.

