# OpenReview forum: "Improving Classifier-Free Guidance in Masked Diffusion: Low-Dim Theoretical Insights with High-Dim Impact"
_ICLR.cc/2026/Conference — ICLR 2026 Poster_

### Official Review · Reviewer_Yvo9 · 2025-10-29

**Soundness:** 3
**Presentation:** 2
**Contribution:** 3
**Rating:** 4
**Confidence:** 2

**Summary:**

This paper investigates Classifier-Free Guidance (CFG) in the context of masked discrete diffusion models. The authors claim to have identified a key flaw in existing CFG implementations, where high guidance strength can lead to imbalanced transitions and unmasking too rapidly, which degrades sample quality.

To address this issue, the authors propose a novel CFG mechanism based on a column normalization of the rate matrix, achievable via a one-line code change. The authors provide a low-dimensional (1D and 2D) theoretical analysis to motivate this change and to characterize the properties of effective guidance schedules.  Experimental results are also provided to validate the technique.

**Strengths:**

1. The paper's primary contribution is a fix for a problem in discrete CFG that can be implemented with a one-line code change. I always like seeing examples of this kind of simple and effective solution, especially when they are theoretically motivated.
2. The methodology of using low-dimensional (1D and 2D) theoretical analysis to derive insights that are relevant to the high-dimensional setting is a strong and interesting approach.
3. Based on the experimental results, the proposed normalization fix appears to be effective.

**Weaknesses:**

My current impression of the paper is that the authors have indeed likely identified a flaw in CFG for masked discrete diffusion models and found a simple fix for it. However, the more serious weaknesses I outline below prevented me from more deeply appreciating the authors' contributions, which has me currently sitting on the negative side of the fence. That said, I admit to being a good deal less familiar with the discrete diffusion literature than I am with the continuous diffusion literature, hence the low confidence in my score. I will keep an open mind during the discussion period, and I look forward to seeing the authors' responses and the feedback from the other reviewers.

I'll list my concerns in decreasing order of seriousness:
1. There seems to be a major contradiction in the paper's claims regarding effective guidance schedules. In Section 3.4 (lines 368-370), the 2D theoretical analysis concludes: "Therefore, effective schedules have higher guidance in the beginning and middle phases of the generation, and their effect towards the end is negligible." This directly contradicts empirical results from numerous other works as well as statements in this paper's abstract and the discussion in Section 4.2: "[S]chedules that apply stronger guidance during the middle and later stages of the sampling process, while keeping early guidance small, tend to perform better." Unless there is a serious misreading on my part, the results of Section 3.4 are never reconciled with these statements and the empirical record.
2. Considering the weight given to the paper's theoretical results, I found them very hard to follow in their current form. Notation is occasionally not defined, and it was sometimes challenging to match the theorems or lemmas with their counterparts and proofs in the appendix.
3. On the topic of notation, in most of the CFG literature, in both continuous and discrete diffusion, $x$ is a state and $y$ is a condition. But in this paper $y$ (never explicitly defined) is apparently a member of the state space (judging by the formula for the partition function) as well as a condition. I found this quite confusing.
4. There are some minor typos and other errors in the paper (e.g. *diffusion* is misspelled in the header for Section 2.1, the regionalism *smoothen* is used in the abstract instead of the standard *smooth*).
5. In the literature, the probability ratio the authors refer to as the *score* is referred to as the *concrete score*. I recommend sticking to this terminology to avoid confusion with the score as it's understood in the continuous diffusion literature.

**Questions:**

1. Can the authors resolve the central contradiction (Weakness #1)? Which finding is correct: the 2D theory suggesting high early guidance, or the empirical results suggesting high late guidance? Why does the theory not seem to match the practice here?

2. In the abstract, what is meant by "late-stage guidance has a larger effect"? Is this a positive or negative effect on generation quality? (This also relates to Weakness/Question #1.)

**Details Of Ethics Concerns:**

No concerns.

---

> ### Author Response · Authors · 2025-11-22
>
> We thank the reviewer for his insightful comments that have allowed us to improve our submission. We went ahead and corrected the typos and added links to the more general versions of the theorems in the appendex as suggested by the reviewer.
>
> > Can the authors resolve the central contradiction (Weakness #1)? Which finding is correct: the 2D theory suggesting high early guidance, or the empirical results suggesting high late guidance? Why does the theory not seem to match the practice here?
>
> We are incredibly thankful to the reviewer for noting this, and apologize for creating this confusion. This was a mistake caused by confusion between the generation time and the forward process diffusion time. The theoretical insights are consistent with what was written elsewhere in the paper. We have corrected this sentence to properly highlight our results and avoid this contradiction.
>
> We should've written "effective schedules have higher guidance in the beginning and middle phases of the diffusion time  ($t \approx 0$)", however noting that having these two times can create confusion (as shown by our own mistake) we have revised it to be consistent with what we wrote in other places "effective schedules have higher guidance at the final and middle phases of the generation while keeping early guidance small".
>
> Once again we would like to thank the reviewer for finding this mistake and allowing us to clarify it.
>
> > On the topic of notation, in most of the CFG literature, in both continuous and discrete diffusion $x$, is a state and $y$ is a condition. But in this paper (never explicitly defined) is apparently a member of the state space (judging by the formula for the partition function) as well as a condition. I found this quite confusing.
>
> We thank the reviewer for allowing us to clarify this. In section 2.2 we introduced the standard setting for CFG using: $$p^{(w)}(x|y) \propto p^w(x|y) p^{1-w}(x)$$
> where $x$ is a member of the state and $y$ is a member of the space as mentioned by the reviewer. However, as we explained further, we move away from this kind of notation since recent works like [1] have shown that to apply guidance it is no longer necessary to consider a conditional and unconditional model, but rather just a target distribution $p$ and a guiding distribution $q$ from which we can use:
> $$p^{(w)}(x) \propto p^w(x) q^{1-w}(x) $$
> Under this setting it is no longer necessary to consider conditions, which allowed us to simplify the notation. We hope that this can clarify our notation, to further imrpvoe this we have changed the equations in line $190$ to only use $x$ for readers more familiar with the standard notation.
>
> [1] Karras, Tero, et al. "Guiding a diffusion model with a bad version of itself." Advances in Neural Information Processing Systems 37 (2024): 52996-53021.
>
> > In the abstract, what is meant by "late-stage guidance has a larger effect"? Is this a positive or negative effect on generation quality? (This also relates to Weakness/Question #1.)
>
> We have revised the abstract to make this more clear as well as fixed other typos mentioned by the reviewer. Guidance towards the end of the generation has a larger positive effect.

---

### Official Review · Reviewer_DWkQ · 2025-10-30

**Soundness:** 3
**Presentation:** 2
**Contribution:** 3
**Rating:** 6
**Confidence:** 4

**Summary:**

This paper analyzes the effect of using guidance schedules in discrete diffusion models and how such schedules can improve performance compared to constant guidance. Drawing inspiration from continuous diffusion models, the authors note that high guidance scales are necessary to enhance sample quality and condition alignment; however, using a constant high guidance scale results in suboptimal performance due to large changes during the early sampling stages. They first demonstrate that transition matrices should be normalized (via SoftMax) in the presence of guidance. Then, using a low-dimensional setup, they show that a constant guidance scale leads to poor performance, reducing diversity and increasing bias in the final generation. To address this, the authors propose an increasing weight schedule to balance the effect of guidance across sampling steps, leading to improved performance across various diffusion models for both image and text generation.

**Strengths:**

- Since guidance in discrete diffusion models is underexplored, this paper bridges an important gap between CFG in continuous diffusion models and discrete diffusion models. The results have potential implications for all systems that rely on discrete diffusion.

- The proposed method is simple and can be easily integrated into existing sampling pipelines.

- The theoretical results provide intuition and reasoning behind the method, although their presentation could be significantly improved.

- Experiments are conducted on both image and text generation benchmarks.

**Weaknesses:**

- In my opinion, the main weakness of the paper is its presentation. Several parameters are either misused in notation or not defined prior to their introduction in the text. This makes the paper difficult to follow and obscures the intuition and analysis behind the proposed method.

- Section 3.4 appears to contradict the main message of the paper. It states that “effective schedules have higher guidance in the beginning and middle phases of generation,” whereas the best performance is reported for “schedules that apply stronger guidance during the middle and later stages of the sampling process, while keeping early guidance small.”

**Minor**:
- There might be a typo in Equation 6.
- Table 1 indicates that the increasing schedule has one parameter, while Table 2 shows it has two parameters ($w$ and $r$).
- The theoretical analysis could also consider the effect of guidance on condition alignment, in addition to its impact on diversity and stability.

**Questions:**

1. Do normalization and time-dependent schedules also improve the performance of Simple Guidance?

2. Could you provide additional metrics, such as Precision and Recall, to separately evaluate diversity and quality, rather than relying solely on FID?

---

> ### Author Response · Authors · 2025-11-22
>
> > In my opinion, the main weakness of the paper is its presentation. Several parameters are either misused in notation or not defined prior to their introduction in the text. This makes the paper difficult to follow and obscures the intuition and analysis behind the proposed method.
>
> We thank the reviewer for their feedback. We have revised the paper to ensure that notation is properly defined. For instance, we found that $\bar{\sigma}_t$ had not been properly introduced. We will continue to revise the manuscript and are open to hearing more valuable feedback on this front.
>
> > Section 3.4 appears to contradict the main message of the paper. It states that “effective schedules have higher guidance in the beginning and middle phases of generation,” whereas the best performance is reported for “schedules that apply stronger guidance during the middle and later stages of the sampling process, while keeping early guidance small.”
>
> We are incredibly thankful to the reviewer for noting this, and apologize for creating this confusion. This was a mistake caused by confusion between the generation time and the forward process diffusion time. The theoretical insights are consistent with what was written elsewhere in the paper. We have corrected this sentence to properly highlight our results and avoid this contradiction.
>
> We should've written "effective schedules have higher guidance in the beginning and middle phases of the diffusion time  ($t \approx 0$)", however noting that having these two times can create confusion (as shown by our own mistake) we have revised it to be consistent with what we wrote in other places "effective schedules have higher guidance at the final and middle phases of the generation while keeping early guidance small".
>
> Once again, we would like to thank the reviewer for finding this mistake and allowing us to clarify it.
>
> > Table 1 indicates that the increasing schedule has one parameter, while Table 2 shows it has two parameters (w and r).
>
> When we refer to an increasing schedule, we mean a simple schedule like $w(1-t)$ as introduced in [1]. The schedules in Table 2 were meant for a more comprehensive empirical study of guidance schedules. Furthermore, since in Figure 7b we found that the optimal value of $r$ corresponds to $r=0$ we find that it is not necessary to tune $r$.
>
> [1] Wang, Xi, et al. "Analysis of Classifier-Free Guidance Weight Schedulers." Transactions on Machine Learning Research Journal (2024).
>
> > Do normalization and time-dependent schedules also improve the performance of Simple Guidance?
>
> This is a very good question, Simple guidance performs interpolation of transition probabilities, while the normalization is for the transition matrices. In this sense it is not possible to apply normalization to simple guidance due to the nature of the mechanism.
>
> Regarding schedules for simple guidance, we did some preliminary tests  and found that the choice of schedule can also be helpful. However, we did not perform a comprehensive study for simple guidance as we found our normalized version to be a superior mechanism.
>
> > Could you provide additional metrics, such as Precision and Recall, to separately evaluate diversity and quality, rather than relying solely on FID?
>
> Thank you very much for the great suggestion. We completely agree, that measuring the tradeoff between sample fidelity and diversity is of great importance for guidance mechanisms. Following [1] we report precision and recall. Precision measures the proportion of generated images that lie close to the real data manifold (fidelity), while recall quantifies the coverage of the real data distribution (diversity), allowing us to understand this correctly.
>
> **Precision/Recall**
> | Method/Strength| w = 1 (no guidance)| w = 2 | w = 3 | w = 4 | w=5 |
> | -------- | -------- | -------- | -------- | -------- | -------- |
> | Our Method   | 0.48 / 0.73|0.51 / 0.75|0.51 / 0.75|0.52 / 0.76|0.18 / 0.72|
> | Unlocking Guidance   | 0.48 / 0.73|0.43 / 0.79|0.29 / 0.77|0.19 / 0.77|0.12 / 0.76|
> | Simple Guidance   | 0.46 / 0.72|0.45 / 0.79|0.34 / 0.76|0.24 / 0.74|0.19 / 0.72|
>
> **Intepretation**
> As guidance strength increases from w=1 to w=5, all methods exhibit  stable recall, ranging from 0.72 to 0.79. The precision statistic is more revealing: for Unlocking and Simple guidance, using $w \geq 1$ always results in degradated precision (worse fidelity) while our method is capable of improving on it (better fidelity). At large guidance scales $w = 5$, all methods suffer from a significant drop in precision. Putting this together, all methods show similar diversity across guidance strengths. However, only our method can improve the fidelity while maintaining diverse samples.
>
> [1] Kynkäänniemi, Tuomas, et al. "Improved precision and recall metric for assessing generative models." Advances in neural information processing systems 32 (2019).

---

### Official Review · Reviewer_xrKo · 2025-11-03

**Soundness:** 3
**Presentation:** 3
**Contribution:** 3
**Rating:** 6
**Confidence:** 2

**Summary:**

The paper analyzes CFG for masked diffusion, showing (in a low-dimensional setting) that strong early guidance hurts quality while late guidance helps. It also identifies an implementation flaw that causes over-eager early unmasking and proposes a one-line fix to smooth the transition from the masked prior to the data distribution.

**Strengths:**

1. CFG is a well researched topic in continuous diffusion. But in masked/discrete diffusion is under active exploration. Clarifying scheduling effects is valuable for both image inpainting/masked modeling and text infilling models
2. Section 3.4 provides exhaustive analysis across factors such as time parameters and guidance strength.
3. The experiment in Section 2.3 is intuitive, improving explainability.
4. The method can be performed in inferencing time.

**Weaknesses:**

1. **Novelty:** The importance of guidance scheduling and rescaling by conditional/unconditional norms has been reported by Kynkäänniemi et al. (2024). Can you clarify how your approach differs?
2. **Metrics/Benchmarks:** For text-to-image evaluation, you only report ImageReward. Could you also evaluate with **HPSv2** to check for aesthetic trade-offs? Additionally, please test on T2I benchmarks like **GenEval** and **T2I-CompBench**.

[1] Kynkäänniemi, T., Aittala, M., Karras, T., Laine, S., Aila, T., & Lehtinen, J. (2024). *Applying guidance in a limited interval improves sample and distribution quality in diffusion models.* NeurIPS 37, 122458–122483.

**Questions:**

1. Prior work reports that higher guidance weights can distort images and reduce fidelity, yet in your Figure 8 larger guidance weights yield better performance. Can you justify this discrepancy?
2. The appendix includes many generated images—could you also include the text prompts used to produce them?
3. Unified reward models are surging. Could you evaluate your method with Show-o [1] and Dual Diffusion [2] to assess its effectiveness in that setting?

[1] Xie, J., Mao, W., Bai, Z., Zhang, D. J., Wang, W., Lin, K. Q., ... & Shou, M. Z. (2024). Show-o: One single transformer to unify multimodal understanding and generation. arXiv preprint arXiv:2408.12528.
[2] Li, Z., Li, H., Shi, Y., Farimani, A. B., Kluger, Y., Yang, L., & Wang, P. (2025). Dual diffusion for unified image generation and understanding. In Proceedings of the Computer Vision and Pattern Recognition Conference (pp. 2779-2790).

---

> ### Author Response · Authors · 2025-11-22
>
> > Novelty: The importance of guidance scheduling and rescaling by conditional/unconditional norms has been reported by Kynkäänniemi et al. (2024). Can you clarify how your approach differs?
>
> We thank the reviewer for allowing us to explain the difference with [1]. [1] is an empirical paper on continuous diffusion, where they demonstrated that applying guidance in a limited interval improves the sampling process. Our work draws inspiration from [1] to try to assess if similar effects will ocurr for discrete diffusion (in contrast to continuous), we perform both an empirical and theoretical study (Our work is the first to analyze guidance schedules theoretically), where we found that similar conclusions follow and furthermore explained why this benefit occurs from a theoretical perspective. As a side benefit of our analysis we discovered that normalization is of great importance to apply CFG in discrete diffusion.
>
>
> [1] Kynkäänniemi, T., Aittala, M., Karras, T., Laine, S., Aila, T., & Lehtinen, J. (2024). Applying guidance in a limited interval improves sample and distribution quality in diffusion models. NeurIPS 37, 122458–122483.
>
> > Prior work reports that higher guidance weights can distort images and reduce fidelity, yet in your Figure 8 larger guidance weights yield better performance. Can you justify this discrepancy?
>
> This is an insightful question, it is true that larger guidance weights can distort images. However, depending on the application how large can vary, traditionally text-to-image applications require higher guidance weights for instance, [1] used $w=9$ and [2] uses values up to $w = 7$ without loss of quality. Other applications may use even larger values for instance [3] used values as high as $200$, although they report a loss in quality at this point. In short the size of the guidance weight depends on the problem. Since we leverage [1] as a pretrained model the values of guidance we required larger values than standard text-to-image continuous diffusion models, but our values are consistent with those used in [1].
>
> [1] Bai, Jinbin, et al. "Meissonic: Revitalizing masked generative transformers for efficient high-resolution text-to-image synthesis." The Thirteenth International Conference on Learning Representations. 2024.
> [2] Chen, Junsong, et al. "Pixart-$\alpha $: Fast training of diffusion transformer for photorealistic text-to-image synthesis." arXiv preprint arXiv:2310.00426 (2023).
> [3] Yang, Xiaofeng, et al. "Text-to-image rectified flow as plug-and-play priors." arXiv preprint arXiv:2406.03293 (2024).
>
>
> > The appendix includes many generated images—could you also include the text prompts used to produce them?
>
> We thank the reviewer for noting this. We went ahead and added these to the paper.
>
> > Unified reward models are surging. Could you evaluate your method with Show-o [1] and Dual Diffusion [2] to assess its effectiveness in that
>
> We are currently working on evaluating on these models. We will update the rebuttal once we have some results.
>
> [1] Xie, J., Mao, W., Bai, Z., Zhang, D. J., Wang, W., Lin, K. Q., ... & Shou, M. Z. (2024). Show-o: One single transformer to unify multimodal understanding and generation. arXiv preprint arXiv:2408.12528.
> [2] Li, Z., Li, H., Shi, Y., Farimani, A. B., Kluger, Y., Yang, L., & Wang, P. (2025). Dual diffusion for unified image generation and understanding. In Proceedings of the Computer Vision and Pattern Recognition Conference (pp. 2779-2790).

---

### Official Review · Reviewer_sPcA · 2025-11-10

**Soundness:** 2
**Presentation:** 2
**Contribution:** 2
**Rating:** 4
**Confidence:** 4

**Summary:**

The paper analyzes classifier-free guidance (CFG) in masked discrete diffusion and shows—theoretically (in low dimensions)—that strong early guidance harms quality, whereas late guidance has a larger positive impact. It identifies a flaw in common CFG implementations, where imbalanced transitions unmask tokens too quickly. To remedy this, the paper proposes a simple one-line column/softmax normalization that smooths transport. Empirically, this normalization improves ImageNet-256 FID, ImageReward text-to-image alignment, and MATH-500 text generation accuracy.

**Strengths:**

The paper presents a principled low-dimensional analysis of CFG for masked discrete diffusion, showing that strong early guidance is harmful while late guidance is beneficial.
Building on this insight, it introduces an elegant, theory-grounded tweak—a one-line column/softmax normalization—that corrects imbalanced early unmasking.
Importantly, by linking this tractable normalization to improved robustness and better FID/ImageReward/MATH-500 results, the paper offers a practical change likely to be widely adopted in discrete diffusion implementations.

**Weaknesses:**

1. The paper develops analysis and closed-form expressions only in 1–2 dimensions for masked discrete diffusion. As a result, guarantees for realistic high-dimensional CTMCs remain implicit, leaving the theoretical treatment somewhat loose.

2. Some results isolate the mechanism using a simple sampler without remasking and with fixed step counts (e.g., 50 steps on ImageNet-256), which may limit generality. Moreover, sampling schedules and samplers are crucial to implementing the guidance mechanism, yet the paper provides neither theoretical analysis nor empirical evaluation of their effects.

3. Although the evidence spans ImageNet-256 FID, ImageReward, and MATH-500 with a single LLM backbone, broader discrete domains (e.g., ASR, text, protein) and larger vocabularies are not explored. Because the approach is closely related to Unlocking Guidance and Simple Guidance, running additional experiments on the datasets used in those works is crucial.

4. Quality gains are reported, but diversity metrics are missing. Because guidance mechanisms can reduce sampling diversity, it is important to report diversity measures and quantify the proposed method’s impact.

5. It is recommended to ensure the citation style is applied uniformly across the manuscript.

**Questions:**

1. Since the theory is developed in 1–2D masked diffusion, could you provide bounds or a proof sketch that extends to realistic high-d CTMCs?

2. With results limited to ImageNet-256 FID and MATH-500 on a single LLM backbone, is there evidence that the approach generalizes to other discrete diffusion tasks?

3. Although normalization is cheap, schedule changes can alter unmasking rates and step counts. Therefore, reporting the resulting overhead (wall-clock, GPU-hours, memory) is important to demonstrate the strength of the proposed method.

4. The proposed framework mainly targets masked diffusion, what changes are needed for uniform or other discrete diffusions?

---

> ### Author Response · Authors · 2025-11-22
>
> > The paper develops analysis and closed-form expressions only in 1–2 dimensions for masked discrete diffusion. As a result, guarantees for realistic high-dimensional CTMCs remain implicit, leaving the theoretical treatment somewhat loose.
>
> We agree that currently our theory relies on low dimensional settings. Although the analysis can be extended to higher-dimensions, the complexity of the problem grows exponentially with the dimension, leading to increasingly intricate expressions and reduced interpretability.
>
> We must also note that this limitation is not unique to our work. For instance, [1] is one of the few existing work on quantitative understanding of classifier-free guidance (CFG), for continuous diffusion and widely acknowledged; it is a pure theoretical paper and yet its analysis is limited to just 1D and for toy distributions only under constant schedules. On the other hand we handle dimension 2 for general distributions under non-constant schedules.
>
> Our work represents a first theoretical step in understanding guidance and its schedule in discrete diffusion models, where analytical results are even more scarce. While our mathematical results are derived in lower dimensions, our goal is to use low-dimensional insights to improve the complicated high dimensional problems, not to provide a complete theory of CFG.
>
> Our analysis revealed structural insights—such as the effects of early versus late guidance and the importance of normalization to balance transitions, and our empirical results demonstrate that our insights generalize to more complex, higher-dimensional practical problems. We hope this work could motivate further theoretical and empirical research to bridge this gap.
>
> [1]: *"What does guidance do? A fine-grained analysis in a simple setting."* Advances in Neural Information Processing Systems 37 (2024): 84968–85005.
>
> > Some results isolate the mechanism using a simple sampler without remasking and with fixed step counts (e.g., 50 steps on ImageNet-256), which may limit generality. Moreover, sampling schedules and samplers are crucial to implementing the guidance mechanism, yet the paper provides neither theoretical analysis nor empirical evaluation of their effects.
>
> Regarding idealized assumptions without analyzing discretization/score-approximation error, our focus on exact scores and exact reverse dynamics follows standard theoretical and practical practices aimed at isolating the intrinsic effect of classifier-free guidance.
>
> From a theoretical point of view discretization error due to different samplers can be incorporated directly: our formulas remain applicable when replacing the exact reverse dynamics with standard numerical solvers such as Euler-Maruyama, and the qualitative behavior we predict persists under practical small step sizes.
>
> From a practical point of view the number of steps and different samplers have been well studied both theoretically and empirically. For instance, [1] performed a comprehensive study of different solvers for discrete diffusion. Despite this, we are currently performing more ablations and will update once we have the results.
>
> [1] Ren, Yinuo, et al. "Fast solvers for discrete diffusion models: Theory and applications of high-order algorithms." arXiv preprint arXiv:2502.00234 (2025).

---

> > ### Author Response · Authors · 2025-11-22
> >
> > > Although the evidence spans ImageNet-256 FID, ImageReward, and MATH-500 with a single LLM backbone, broader discrete domains (e.g., ASR, text, protein) and larger vocabularies are not explored. Because the approach is closely related to Unlocking Guidance and Simple Guidance, running additional experiments on the datasets used in those works is crucial,.., is there evidence that the approach generalizes to other discrete diffusion tasks?
> >
> > We must respectfully note that the methods proposed in our paper, as well as Unlocking and Simple Guidance are agnostic to the dataset and architecture and are just based on the fundamental mathematical process of sampling from discrete diffusions. For this reason testing on every single application is not strictly necessary. Our current experiments already include large vocabularies, for instance, the MATH-500 benchmark has a vocabulary of more than $120$K tokens. Despite this in order to gain the reviewers favor, we leverage the codebase from [1] and train on QM9 using masked diffusion, QM9 was one of the datasets used in [1] to evaluate guidance and it is a molecule dataset. We present the results of this experiment in Appendix F. In our short, we found that normalization is also helpful in this domain.
> >
> > [1] Schiff, Yair, et al. "Simple guidance mechanisms for discrete diffusion models." arXiv preprint arXiv:2412.10193 (2024).
> >
> > > Quality gains are reported, but diversity metrics are missing. Because guidance mechanisms can reduce sampling diversity, it is important to report diversity measures and quantify the proposed method’s impact.
> >
> > Thank you very much for the great suggestion. Measuring the tradeoff between sample fidelity and diversity is of great importance for guidance mechanisms. Following [1] we report precision and recall. Precision measures the proportion of generated images that lie close to the real data manifold (fidelity), while recall quantifies the coverage of the real data distribution (diversity), allowing us to understand this correctly.
> >
> > **Precision/Recall**
> > | Method/Strength| w = 1 (no guidance)| w = 2 | w = 3 | w = 4 | w=5 |
> > | -------- | -------- | -------- | -------- | -------- | -------- |
> > | Our Method   | 0.48 / 0.73|0.51 / 0.75|0.51 / 0.75|0.52 / 0.76|0.18 / 0.72|
> > | Unlocking Guidance   | 0.48 / 0.73|0.43 / 0.79|0.29 / 0.77|0.19 / 0.77|0.12 / 0.76|
> > | Simple Guidance   | 0.46 / 0.72|0.45 / 0.79|0.34 / 0.76|0.24 / 0.74|0.19 / 0.72|
> >
> > **Intepretation**
> > As guidance strength increases from w=1 to w=5, all methods exhibit stable recall, ranging from 0.72 to 0.79. The precision statistic is more revealing: for Unlocking and Simple guidance, using $w \geq 1$ always results in degradated precision (worse fidelity) while our method is capable of improving on it (better fidelity). Putting this together, all methods show similar diversity across guidance strengths. However, only our method can improve the fidelity while maintaining diverse samples.
> >
> > [1] Kynkäänniemi, Tuomas, et al. "Improved precision and recall metric for assessing generative models." Advances in neural information processing systems 32 (2019).
> >
> > > Although normalization is cheap, schedule changes can alter unmasking rates and step counts. Therefore, reporting the resulting overhead (wall-clock, GPU-hours, memory) is important to demonstrate the strength of the proposed method.
> >
> > Unfortunately, we did not measure these quantities when performing the experiments in the paper. We run the following experiment to get a more complete understanding of the extra computation added by normalization. We generate $5000$ images using a batch size of $125$ using 4 NVIDIA A100 GPUs using Halton-MaskGIT under a constant guidance schedule with $w = 2$.
> >
> > | Method| Time (Minutes) | GPU Hours | GPU Memory (GB)|
> > | ----- | -------------- | --------  | -------- |
> > | Our Method   |11.91|0.79| 75.2 |
> > | Unlocking Guidance   |11.91|0.79| 75.2 |
> > | Simple Guidance   |11.93|0.79| 75.2 |
> >
> > We see that all methods are equally computationally expensive and that no overhead has been introduced by normalizing.
> >
> > > It is recommended to ensure the citation style is applied uniformly across the manuscript.
> >
> > We have revised the submission to ensure a consistent citation style. Following ICLR guidelines we used \citet for in text citations and \citep for other cases. We hope that this can help improve our submission, and we are thankful for the feedback.
> >
> > > The proposed framework mainly targets masked diffusion, what changes are needed for uniform or other discrete diffusions?
> >
> > We thank the reviewer for asking this insightful question. In appendix E we present how to implement the normalization for a general diffusion process as well as a small experiment for uniform diffusion on QM9 following the experimental setup in the Simple Guidance Mechanisms paper. We found the results on QM9 to be harder to interpret, but generally found that normalization was never harmful and provided more robustness to the guidance scale.

---

### Author Response · Authors · 2025-12-01

We thank the reviewers for their time and helpful comments that have allowed us to improve our submission. After the revision we have incorporated the following

1. Included new experiments on QM9 for masked diffusion to test on more diverse datasets
2. Included new experiments using Show-o as a demonstration of our guidance for mixed models
3. Included an analysis of precision/recall for fidelity/diversity tradeoffs

We have added these new results in the appendix and hope that these new additions can improve the quality of our submission and are grateful for the constructive criticism.

Sincerely,
The Authors

---

### Meta-Review · Area_Chair_upV5 · 2026-01-07

**Summary:**

This paper presents an analysis of classifier-free guidance in the context of masked diffusion models, revealing that strong early guidance hurts quality while late guidance helps, particularly in a low-dimensional setting. Reviewers xrKo and DWkQ provided positive assessments. Reviewer Yvo9 raised concerns about the guidance schedule and the writing, and reviewer sPcA noted some limitations in the theoreteical result. The authors have satisfactorily addressed some of these concerns through their rebuttal. As noted by reviewer xrKo, the topic of guidance  in masked/discrete diffusion  is under active exploration. This paper offers a valuable, focused contribution to that discourse, and it is generally sound and novel. Thus I recommend acceptance.

**Reviewer Concerns:**

Most reviewer concerns have heen addressed by the rebuttal, such as concerns regarding clarify about motivation and the analysis of efficiency.

**Reviewer Scores:**

There is no clear evidence to suggest that any reviewer would have changed their score after a full discussion.

---

### Decision · Program_Chairs · 2026-01-26

Accept (Poster)